# Retrofitting Cost Modeling in Aircraft Design

**Pierluigi Della Vecchia \*** , **Massimo Mandorino, Vincenzo Cusati and Fabrizio Nicolosi**

DAF Research Group, Department of Industrial Engineering, University of Naples Federico II, Via Claudio 21, 80125 Naples, Italy; massimo.mandorino@unina.it (M.M.); vincenzo.cusati@unina.it (V.C.); fabrizio.nicolosi@unina.it (F.N.)

**\*** Correspondence: pierluigi.dellavecchia@unina.it

**Abstract:** Aircraft retrofitting is a challenging task involving multiple scenarios and stakeholders. Providing a strategy to retrofit an existing platform needs detailed knowledge of multiple aspects, ranging from aircraft performance and emissions, development and conversion costs to the projected operating costs. This paper proposes a methodology to account for retrofitting costs at an industrial level, explaining the activities related to such a process. Costs are mainly derived from three contributions: development costs, conversion costs and equipment acquisition costs. Different retrofitting packages, such as engine conversion and onboard systems electrification, are applied in the retrofitting of an existing 90 PAX regional turbofan aircraft, highlighting the impact on both aircraft performance and industrial costs. Multiple variables and scenarios are considered regarding trade-offs and decision-making, including the number of aircraft to be retrofitted, the heritage of an aircraft and its utilization, the fuel price and the airport charges. The results show that a reduction of 15% in fuel demand and emissions are achievable, considering a fleet of 500 platforms, through a conspicuous investment of around EUR 20 million per aircraft (50% of the estimated price). Furthermore, depending on the scenarios driven by the regulatory authorities, governments or airlines, this paper provides a useful methodology to evaluate the feasibility of retrofitting activities.

**Keywords:** retrofitting costs; aircraft design; collaborative design; MDO





## 1. Introduction

The aviation industry must face increasingly hard challenges owing to new regulations, constraints and customer needs. Nowadays, airlines must comply with stricter noise and emission rules, safety demands, passengers' increasing expectations of comfort and fuel price volatility. Some of these challenges have led to the birth of different countries' government-funded development programs, which are attempting to support advances in research and technology. One of the most important targets of these programs is a reduction in the environmental impact of air transport, in combination with improvements in mobility and passenger satisfaction, cost efficiency, safety and security [1]. The aviation ecosystem (including airlines, aerospace manufacturers, airports, air navigation service providers and business aviation) have reached an agreement under which carbon dioxide ($CO_2$) aviation emissions must be reduced by 50% with reference to the year 2005, no later than 2050. One of the strategies to be adopted comprises the deployment of innovative technology [2].

Typically, a new aircraft generation is brought in to replace older models in the same seat category every 15 to 20 years or so. The introduction of many new models in the current period (2014–2020) might result in an innovation gap in the second half of the 2020s before demand for a follow-on for the current new aircraft generation will arise. This could lead to a noticeable slowdown in the average fuel efficiency improvement [3].

In such a situation, an attractive solution could be the upgrade of some aircraft components. Engine replacement, advances in materials and aerodynamic improvements could lead to benefits in terms of fuel consumption and emissions, without necessitating the

amount of time required to introduce a new aircraft generation. A fleet of aircraft could take advantage of these technologies, satisfying strict regulations, guaranteeing more comfort for passengers and improving company reputations. A significant example is represented by aircraft interior updates. Typically, the cabin interior is characterized by a maintenance cycle at different times from those of the aircraft's overhaul and maintenance schedules. Usually, airlines retrofit their cabin products every 5–7 years [4], which is a very brief time range if compared with other updates. As a matter of fact, the principal international original equipment manufacturer (OEM), Airbus, Boeing and Embraer declared that they spent only around 6 years on advanced and detailed design, manufacturing and certification phases before launching the A320Neo, the 737MAX deliveries and the E2 family [5,6].

The most crucial point of a retrofitting operation consists of its commercial applicability [7]. The return on investment (ROI) represents an important aspect to be considered, but it is not necessarily the main motivation that leads an OEM to undertake the retrofitting activity.

During the last few decades, the market potential for retrofitting activity has continuously grown, generating different opportunities. Indeed, as stated in the RETROFIT research project [8], retrofitting is not necessarily aimed at aged and/or out-of-production aircraft only. The retrofit activity does not necessarily include only maintenance repair and overhaul (MRO) activities, considering that these activities consist of aircraft modification without implementing innovative components. Nevertheless, the OEM could be a significant contributor since it can provide certified data, help with certification and configuration management, and reach a large share of customers. A good representation of this phenomenon is provided by the programs known as "Performance Improvement Packages" (PIPs), generally conducted by the engine OEM, airframe OEM, or both. Different programs are being developed to deploy technical improvement packages, to be exploited on the most popular in-service engines. These programs are known as "re-engining" programs since they do not usually require major structural modifications to the aircraft. Other possible PIPs programs aim to improve aircraft aerodynamic performance or minimize the time spent on certain operations, such as the access to airports. Performance Improvement Packages have been developed by such Airframe OEMs as Airbus, Boeing and Bombardier [8].

The most appropriate demonstration of the above considerations concerns the "re-engining" activity carried out by Embraer on the E-Jet family [5].

Embraer declares that the performance and economical characteristics are improved with the introduction of this new fleet; the new family of aircraft can carry a higher number of passengers on board. The E2 family has improved operating costs over its predecessors. The cost per aircraft per mile and the maintenance cost per flight go down from EUR 8.0 and EUR 733.6 to EUR 7.54 and EUR 660.3, respectively. The reduction in maintenance cost is obtained thanks to the new engine technology, longer intervals between maintenance, improved system reliability and lower prices for reserve components [5].

Another example of retrofitting activities is offered by the RETROFIT research project [8]. Three possible retrofitting activities were analyzed therein in a qualitative manner: SESAR-compatible avionics installation, engine replacement on an A320 aircraft and the introduction of electric (wheel-driven) taxiing systems. Specifically, based on publicly available data [9,10], the economic impact of a "re-engining" operation applied to the A320 family of aircraft in service (the A319, A320 and A321) was examined. The study was carried out regarding the installation of a Pratt & Whitney geared turbofan engine in aircraft up to 10 years of age. The costs related to conversion design, equipment acquisition, downtime and profit were considered, along with the economic benefits deriving from reductions in fuel demand, emissions and noise-related costs. Based on these considerations, the total capital costs required to update the aircraft were estimated at between EUR 0.8 and EUR 1.1 million per year per aircraft. Instead, the corresponding economic benefit estimation was equal to EUR 0.66 million saved per year per aircraft, showing a non-viable business case for the operators. The RETROFIT project declared that to obtain an appreciable ROI,

the fuel efficiency improvement should be higher than 20% with respect to the previous engine models (the engine considered resulted in 12% of fuel savings). However, what was proposed in the RETROFIT project is not based on quantitative analyses of costs and performance or on assumptions or a priori scenarios that are not always supported by industrial experience: an example is represented by the time needed for re-engining an A320-series aircraft, wherein only 3 weeks of activity was assumed. Moreover, no details are clarified on airport charges (especially on noise) and aircraft maintenance improvements.

In this work, a detailed retrofit with costs methodology analysis is proposed and applied to a 90-passenger turbofan aircraft. The proposed procedure extends the approach introduced in [11], detailing an engine and on-board-system (OBS) retrofitting activities. Three main phases are considered in the process: the development, conversion and equipment acquisition phases. For each one of these steps, recurring and non-recurring costs are evaluated, taking into consideration the possible agreements, discounts, the intellectual and manual operations to be performed, and stakeholders' profits. For each activity, the number of employers and the expected time needed are considered. The cost estimation of each retrofitting sub-phase brings much more accurate results and flexibility, allowing any form of retrofitting costs evaluation. Indeed, it is possible to add or remove some of these sub-phases to analyze a different aircraft upgrade scenario. Conversely, such an approach requires a deep knowledge of the whole process, and it is oriented to an industrial level.

The overall retrofitting costs estimation is applied in the framework of the AGILE 4.0 research project [12], wherein collaborative multidisciplinary aircraft design and optimization are carried out, focusing on the industrial domain of production, maintenance and certification, with the help of multiple industrial partners. Once the overall workflow is assembled, the retrofitting costs (following capital costs) can be compared with the savings (part of the cash DOC contribution coming from fuel, emissions charges and maintenance costs) on different operative scenarios.

The paper is structured as follows. Section 2 provides a description of certain retrofitting activities, illustrating the main benefits and risks. Section 3 describes a general methodology that is useful for estimating the costs due to a retrofitting activity, in particular the costs associated with a single aircraft of the upgraded fleet. A method to account for costs deriving from development, conversion and equipment is explained. In Section 4, the methodology is applied on a reference aircraft, like the Embraer 175, which supports the design stage in the AGILE 4.0 research project. Different kinds of upgrade activities are considered, one related to aircraft re-engining and the second one to an OBS architecture upgrade. In Section 5, the retrofitting propagation effects at the aircraft level are evaluated, carefully considering the masses, performance, emissions and cost variations during the whole design process. Finally, in Section 6, the conclusions and the final remarks of this work are presented, comparing different scenarios.

## 2. Retrofitting Solutions: Benefits and Risks

A wide variety of aircraft components could be retrofitted or upgraded. For each of them, different challenges could be encountered, also leading to different benefits. A detailed description of the most attractive retrofitting activities, including their benefits, risks and applications is provided elsewhere [13]. In summary, retrofitting activities can be divided into three main branches: aerodynamics, engine upgrade and onboard-systems upgrade. In terms of aerodynamics, innovative design solutions have been realized over the past decades to dramatically reduce the amount of drag during mission phases. The introduction of a variable camber, riblets, raked or spiroid wingtips and a system capable of keeping laminar flow on the wing can bring the total fuel consumption reduction from 1 to 10% [3]. One of the most recent examples concerns a flight test campaign performed after replacing the outboard A340 wing with approximately 10-m-long laminar wing panels. This modification can potentially reduce the fuel burned by as much as 4.6% on an 800-nautical-mile trip [14]. System innovations can also lead to a reduction in fuel consumption. For instance, during the taxi phases, an electric system would allow aircraft displacement

without using the main engines, resulting in a 3% fuel saving [3]. In general, systems upgrading operations are related to adjustable landing gear, the introduction of advanced fly-by-wire systems or a structural-health-monitoring system. These innovations can lead to a fuel consumption reduction that can reduce from 1 to 5% of the total fuel burned [3].

Probably the most significant benefit in fuel consumption is offered by new engine technologies. Fan component improvements, the introduction of an advanced combustor, or adaptive flow control can save up to 20% of the fuel burned during a typical mission. Research programs in the Clean Sky and Clean Sky 2 projects, related to propulsion technologies, have developed innovative engines for next-generation aircraft. Geared turbofans, ultra-, and very high bypass ratio architectures allow a significant decrease in fuel consumption, noise and emissions. These new engine architectures allow engineers to achieve up to 25% of $CO_2$ emission reduction compared to previous engines [15]. The most common engine upgrade, which has already been entered into service for several aircraft categories, is represented by high-bypass-ratio (BPR) engine installation. In the field of regional jet aircraft, some representative examples are the MRJ, the Embraer E2 family, and the A220 (formerly C series), A320neo and 737MAX. The engines installed on these aircraft are characterized by 9 to 12 BPR and enable a saving of up to 15% of fuel compared to previous engines with a BPR that is typically 5 or 6. Innovative engines mounted on aircraft such as the A330neo and 777-9 allow engineers to reduce the fuel burn by about 10% [16].

The government and the aviation industry have worked to try and reduce the impact of noise in different ways. Some examples are the promotion of the use of quieter aircraft, restrictions on times during which the airports can operate and the routes on which they can be used. In some cases, the total number of flights that can depart from and arrive at an airport has been limited. Engine retrofitting allows for dramatic noise reduction (along with a noise charge reduction). The Department of Aerospace Engineering of the Delft University of Technology developed a report concerning the aircraft noise and cost impact of a re-engineering operation [17]. The main results are represented by the effect that noise optimization, due to engine replacement, has on the direct operating costs (DOC). The analysis is carried out on the basis of aircraft data related to the Boeing 747, A330 and A320. The rise in direct operating costs (DOC) produced by an engine replacement operation that is performed for reducing aircraft noise by 1 EPNdB corresponds to 0.75% for the flyover, 1.5% for the sideline and 3.0% for the approach phases. This trend is valid up to a reduction of 2–3 dB. However, the EASA certification noise levels, approved by EASA as part of the aircraft certification process, in compliance with the applicable noise standards, as defined in International Civil Aviation Organization (ICAO) Annex 16, Volume I, show that in the category of regional jets, for a maximum take-off weight of from 40 to 80 tons, certified aircraft with an advanced engine can achieve an average cumulative noise reduction of about 8–10 EPNdB [18]. This huge improvement is mainly due to new engine technologies with a higher BPR and to advanced materials. Of course, different solutions can be considered for reducing the noise emissions generated by the engine. Some examples concern the addition of chevrons [19] or fluidic injection [20], which have the advantage of only being applied during the approach and when taking off. Moreover, local restrictions on noise and emissions are becoming progressively more common, imposing operative limitations and surcharges on heritage aircraft in several airports distributed worldwide [21]. Typically, a quieter engine is more expensive than a traditional one (engine acquisition cost). However, the engine price is not a factor that generally influences the engine choice since the prices of all the possible engines to be acquired tend to match during negotiations. What mainly affect the engine choice are the fuel and noise emissions [17] and, of course, fuel flow. Moreover, based on the airport charges, which are set locally and are subject to rapid changes, the impact of retrofitting could be drastically different.

Another interesting retrofitting activity is related to the aircraft onboard system (OBS). The aviation industry is embracing the concept of the more electric aircraft (MEA), to eliminate the many drawbacks suffered by bleed air systems and hydraulic systems (low efficiency, lack of reliability and high maintenance costs) and reduce the carbon footprint

that these systems have generated over the years. In MEA, not all the onboard systems are electrically powered, but most of them are. Some hydraulic, pneumatic and mechanical power sources could still be present. Boeing chose to adhere to the MEA (more electric aircraft) concept with the Boeing 787 Dreamliner. One key change from traditional airliners is the electrical architecture of the 787's flight system; in particular, the BOEING 787 is the first aircraft that adopted a bleedless configuration.

### 3. Retrofitting Costs Methodology

In this paper, a retrofitting costs estimation methodology is proposed. The purpose is to investigate if these potential retrofits could be attractive for the existing fleet of jet airliners. The method allows us to determine the modification costs associated with a single aircraft and, consequently, the upgraded aircraft sale price at the fleet level. These outputs are necessary to estimate the capital costs and expenditures of the modified aircraft, allowing a cost/benefit trade-off.

Generally, aircraft unit cost is mainly divided into recurring costs (RC) and non-recurring costs (NRC). A definition of both categories is provided below.

- Recurring costs. This category includes the costs of all the activities, items and materials that are directly related to the number of aircraft produced. According to Beltramo et al. [22], this includes the costs accrued for all labor and raw materials related to the production of major components and subassemblies by the aircraft manufacturer, for the acquisition of those components that are not produced by the aircraft manufacturer, and for all the labor required to integrate the major components and subassemblies into a finished aircraft.
- Non-recurring costs. This category is mainly linked to the development costs. According to Markish [23], it includes all the operations required to bring an aircraft concept to production. As a consequence, this cost is not dependent on the number of aircraft produced. Non-recurring costs include costs related to the preliminary design, detail design, tooling, testing and certification. All these operations must be considered for each of the parts comprising the aircraft.

Retrofitting costs can also attain a similar distinction, where the final cost of a retrofit per aircraft is the sum of the three major costs listed below (see also Equation (1)).

- Development costs. This coincides with the non-recurring costs, the initial investment to support the retrofit project.
- Conversion costs. This corresponds mainly to the recurring costs. This cost item is associated with the practical actions needed to modify the aircraft.
- Equipment costs. Expenditures on the purchase of every kind of equipment (i.e., engine or OBS), materials and ground equipment correspond to a recurring cost.

$$C_{RETROFIT} = C_{DEV} + C_{CONV} + C_{EQUIP} \tag{1}$$

After the conceptual stage of a new product, the OEM develops a theoretical break-even point wherein the sale of a sufficient number of units has covered the investment in terms of capacity (covering the NRC); conversely, the operators' (airlines') break-even point must show a return on the investment during the time of operation of the product (covering the unit cost). The stakeholders' profit margin should also be accounted for.

As stated by Curran et al. [24], there is no consolidating costs theory upon which the models are based. They individuate several methods, summarized into classic methods, wherein attaining the analogous costing methods (case-based reasoning (CBR)), parametric costing methods (cost-estimating relationships (CER)), bottom-up methods; and advanced methods that include the feature-based method, the fuzzy logic method, the neural network method, and more. Further specific functional classifications are then provided: (1) computational costing and aggregating various identified costs; and (2) relational costing in the comparative relationship of product-defining parameters. Another possible classification is related to causation. Non-causal models can simulate general trends; they are suitable

for understanding systems behavior or for estimating the cost of a complex product. The casual approach is more complete and helps to show what factors drive the output and to formulate the guiding principles.

In general, with reference to the current state of the art, it is possible to state that usually, a substantial effort is made by industries to estimate the specific costs of operations or components instead of developing a global cost-modeling method that can consider the cost of each component in the entire process. The causality of costs and parameters are not considered; in this way, only a specific approach can be developed. This is the reason why cost estimation is typically based on the experienced situation and not on a scientific method. Traditional methods that are useful to estimate design, development, certification and operational costs are based on a parametric equation developed from statistical data. The reports developed by Beltramo et al. [22] and Large at al. [25], respectively, for NASA and the Assistant Secretary of Defense, are examples of possible methods that can be exploited to determine the cost-estimating relationship (CER) for commercial and military transport aircraft. The derived equations relate the economic aircraft characteristic to the components of weight, speed and cruise altitude. A problem related to this kind of relationship-based calculation is that they can be outdated and not suited to a new industrial environment. Roy et al. [26] showed a method for generating an actualized cost-estimating relationship (CER) that is related to the engineering design phase. The main innovation consists in the introduction of quantitative and qualitative knowledge for CER development. Quantitative elements are parameters that can be precisely measured and directly related to the cost. Another example of a qualitative element to consider is the time spent by the designer on thinking about the design problem. An estimation of this related cost can only be made based on industry knowledge and experience.

The proposed methodology aims to provide a detailed process by which to estimate the RC, NRC and equipment costs in a retrofitting process. It is based on an industrial expert's experience and knowledge, focusing on the causal cost relationships and using bottom-up classical methods. As previously mentioned, the basis of the method consists of determining a value for the three costs represented in Equation (1). This estimation is made using the number of hours and the corresponding costs that the development and conversion activities will require during the aircraft update, and via the cost of each piece of equipment needed. A good way to develop these estimations consists of decomposing each one of the three cost groups introduced in Equation (1). Development, conversion and equipment costs can be broken down into multiple sub-categories, as represented in Figure 1. Each sub-group can be further divided according to the specific activities that characterize the retrofitting work under consideration. For instance, development costs include activities such as structural analysis; in this sub-group, flutter analysis will certainly be included if the retrofitting activity that is analyzed consists of a wing structure upgrade. In contrast, a structural dynamic analysis will probably not be included if the retrofitting activity consists of a cabin update. In a similar manner, all the costs related to the sub-groups illustrated in Figure 1 can be considered as being zero or as composed of different activities. A more detailed description of how to compute the sub-categories illustrated in Figure 1 is presented below.

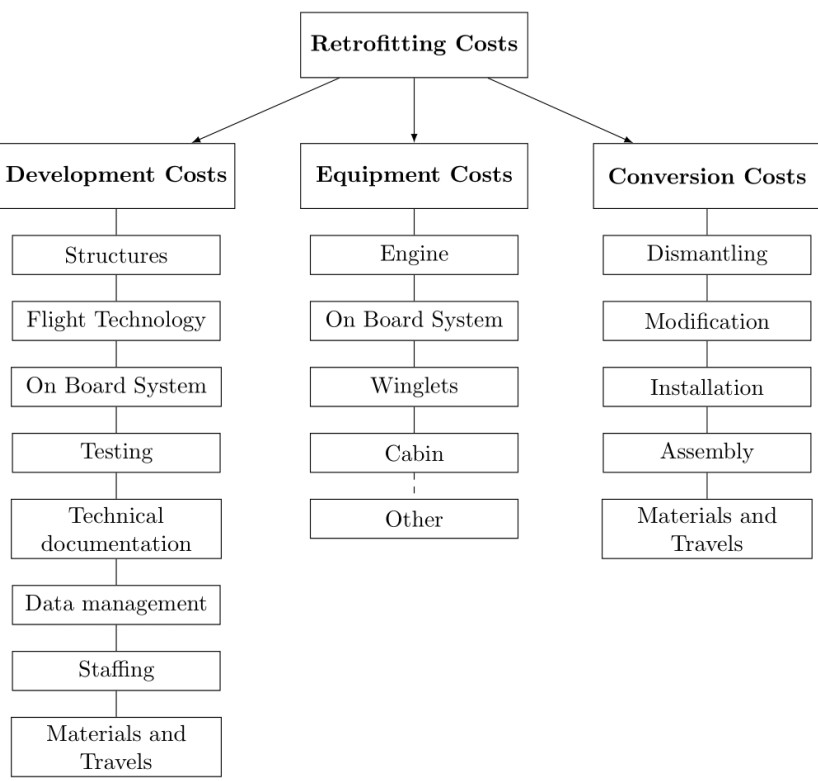

**Figure 1.** General representation of the retrofitting cost breakdown.

*3.1. Development Costs*

The aircraft OEM will require an initial investment in activities related to design and preparing for conversion. This is a capital expenditure that is sustained only once, at the beginning of the work activity life cycle. This category includes, for example, the costs of producing flight manuals, research, testing and all types of engineering efforts associated with aircraft modification.

The logic behind the methodology consists of determining the total development cost by starting from the activities required, establishing the time required to accomplish them in hours and calculating the cost of each activity, assuming these hourly costs. To account for all possible development costs, it is fundamental to consider all the activities required to design the aircraft upgrade. Studies on aircraft structure, flight technology and onboard systems may be required to compute the feasibility of aircraft retrofitting. The costs of wind tunnel tests, in-flight testing, and subsequent analyses must be considered as an additional expenditure, due to the certification activity that must be accomplished to establish aircraft safety airworthiness. Technical documentation, such as repair manuals, an airplane flight manual (AFM) or flight crew operating manual (FCOM) could need to be drafted again or updated; this activity implies more time and costs. Other expenses that should be considered are related to data management, materials acquisition, travel and expenditures for other engineering activities not yet considered (included in staffing): for instance, safety and chief engineers, design quality assurance, costs and planning. By multiplying the total number of hours by the hourly cost, the result is the total development cost of carrying out the aircraft modification. Equation (2) summarizes in a general manner how the total development cost can be computed:

$$C_{DEV} = Ch_E \sum_{i=1}^{N} (Eh_i \cdot Ep_i) + \sum_{j=1}^{M} \left( Ch_{T_j} \cdot Th_j \right) + C_{ODEV}. \tag{2}$$

$Eh_i$ and $Th_j$ represent the hours needed, respectively, for each engineering and testing activity; $Eh_i$ indicates the engineering hours for one employee; this number is then multiplied by $Ep_i$, the number of people needed for that activity. If $Eh_i$ is expressed in months of

a year, the related number of hours to be considered could be 160 in a month and 1760 in a year, as suggested by the LEONARDO company [27]. $Th_j$ refers to the total time required to perform the testing activity considered, without taking into account the number of people required. $N$ is the number of engineering operations needed, including some items in Figure 1 (retrofit design, technical documentation and data management); consequently, index $i$ represents the $i$th engineering operation considered. Analogously, M represents the number of test and certification activities that are required to perform the aircraft upgrade (Testing and Certification, see Figure 1); consequently, index j represents the $j$th testing operation considered. $Ch_E$ and $Ch_{Tj}$ represent, respectively, the hourly personnel cost for engineering and for specific tests that must be performed (for instance, wind tunnel or flight tests). $C_{ODEV}$ represents all other costs associated with complementary development activities or those expenses for which it makes no sense to specify an operational time (Materials and Travels, see Figure 1). For these kinds of activities, only a total cost must be specified. Equation (3) illustrates how to compute $C_{ODEV}$:

$$C_{ODEV} = C_{DOC} + C_{DATA} + C_{STAF} + C_{TRDEV} . \tag{3}$$

$C_{DOC}$ is the cost of documentation, and $C_{DATA}$, the cost for data management, while $C_{STAF}$ represents the cost of all the people not considered in the previous items and $C_{TRDEV}$ represents the travel costs. All these costs can be indicated as a total cost or as the product between the number of people and the hours required for that item. Equation (4) represents a parametric way to estimate the travel costs. In this phase, people's movements are mainly due to engineering activities. This is the reason why the cost of travel can be considered to be dependent on the total hours already estimated for engineering activities and is obtained by multiplying it by a set factor. The value considered for this parameter is EUR 2.25/h per worker, as suggested by the LEONARDO company:

$$C_{TRDEV} = 2.25\sum_{i=1}^{N}(Eh_i \cdot Ep_i) . \tag{4}$$

As can be seen, this method gives the possibility of inserting as an input the hourly cost of every kind of engineering and testing activity. Certain data, such as the engineering cost [28], are strongly correlated to the context in which the activity is performed. A value of EUR 72/h can be considered for countries with a high cost of living. This value can easily be reduced to EUR 36/h for less-developed countries. However, an average value corresponds to EUR 63/h. Other typical data for hourly costs that can be used in Equation (4) are EUR 4500/h for the wind tunnel test [29] and EUR 7500/h for flight tests, as suggested by the LEONARDO company.

### 3.2. Conversion Costs

The aim of this section is to show the estimation of recurring costs from the practical operations to be executed on the aircraft. If the retrofit program consists of the installation of more advanced and fuel-efficient engines, tasks such as the disassembly of old engines and modifications of the wing structure fall into this category. In terms of aircraft maintenance, repair and overhaul (MRO), companies may perform these tasks in dedicated facilities. However, in terms of other aspects, such as certification or testing, the OEM will usually perform the retrofitting.

This methodology correlates every practical activity with the time necessary to carry it out, typically expressed in months and then converted into hours. By multiplying the number of hours by the number of workers involved and the hourly labor rate, the result obtained is the total conversion cost in manpower to complete the aircraft improvement.

Equation (5) represents how the total cost related to all the conversion activity can be estimated:

$$C_{CONV1} = Ch_O\sum_{k=1}^{P}(Np_k \cdot Oh_k) + C_{OCONV} . \tag{5}$$

$Np_k$ and $Oh_k$ represent the number of workers and the time required to perform a single conversion activity by one employee. $P$ is the number of operations considered; this means that the index $k$ represents the $k$th operation that is accounted for. $Ch_O$ is the operations' hourly cost. According to the Bureau of Labor Statistics [30], this cost has increased over the years, reaching its maximum in 2021. The value proposed for the current work is EUR 80/h. This number takes into consideration the manufacturing operations cost required to procure the tools related to these activities. In this case, $C_{OCONV}$ represents all the other costs associated with complementary conversion activities or with expenses for which it makes no sense to specify an operational time (Materials and Travels, see Figure 1). Equation (6) demonstrates how to compute these other costs ($C_{OCONV}$):

$$C_{OCONV} = C_{MAT} + C_{TRCONV} + C_{RDP} . \tag{6}$$

The item presented in Equation (6) can be computed in the same way as already described for $C_{TRDEVO}$ in Equation (4). $C_{MAT}$ is the cost of the materials; this can be computed by multiplying the factor $\sum_{i=1}^{P}(Np_k \cdot Oh_k)$ with EUR 24/h per worker. In the same manner as $C_{TRCONV}$, the cost of traveling can be obtained by multiplying the same factor by EUR 8.0/h. $C_{RDP}$, which accounts for aircraft reception, painting and delivery can be computed by multiplying the previously mentioned summation by EUR 4.0/h. Conversion costs, unlike development costs, are expenses that must be sustained for the activities that take place on every single aircraft. This method employs a learning curve to display these costs, considering the phenomenon of increasing productivity as the number of items produced accumulates. The model adopted in this case is the cumulative average model, as proposed by the authors of [22]. Using Equation (7), it is possible to estimate the cost of every single aircraft, considering the experience garnered after the production of all the previous items. $C_{CONV1}$ is the value obtained via Equation (5) and represents the conversion cost of the first aircraft produced. This cost is not affected by the learning curve. However, $C_{CONVX}$ is the cost of the $X$th aircraft created:

$$C_{CONV_X} = C_{CONV_1} \cdot X^{\log(LR)/\log(2)} . \tag{7}$$

$LR$ represents the learning rate. This is the decline in unit cost obtained after each doubling of cumulative volume. For instance, $LR = 0.8$ means that there is a 20% decrease in average cost each time the cumulative quantity produced is doubled. The value of the learning rate depends on the manufacturer's capability. The value suggested by Beltramo varies between 0.85 and 0.95. The final cost $C_{CONV}$ to be considered for the item's conversion operations, referring to a single aircraft, is the mean value between all the costs considered, according to the learning curve, as represented by Equation (8), where $P$ is the number of operations considered:

$$C_{CONV} = \frac{\sum_{X=1}^{P} C_{CONVX}}{P} . \tag{8}$$

### 3.3. Equipment Costs

The third cost item considered by the methodology to determine the total cost of a retrofit program is that of equipment. The aircraft manufacturer will need to pay for the purchase of equipment to carry out the retrofit. For example, in the case of the addition of winglets in outsourcing, it is necessary to buy them, as well as to buy new onboard systems in the case of an onboard systems upgrade.

Usually, equipment cost is the most difficult one to estimate: the purchase price of each aircraft component is confidential data. Beltramo [22], Sforza [31] and Jenkinson [32] offer only some examples of available methodologies that are able to provide an estimation of all aircraft components' costs via one simple parameter, such as their weight. In certain cases, this kind of estimation is not particularly accurate. This can happen because the methodology is not actualized regarding new technologies. For instance, the price of an innovative engine is different from that of one of the manufacturer's previous versions,

despite the fact that some engine characteristics (thrust, weight, dimensions) are the same. This means that two kinds of engines, characterized by the same static thrust, may be significantly different prices. In this case, the equipment cost estimation should be corrected, keeping in mind the public prices and statistical data, as has been achieved in the following paragraph.

In addition, a discount from an equipment manufacturer to the OEM that retrofits the aircraft can be considered. This type of agreement usually depends on the number of items ordered [33]. The presented methodology provides the possibility of considering a reduction in the equipment acquisition costs by simulating a possible agreement between manufacturer and supplier. Two parameters are defined to build the rule that regulates the manufacturer–supplier agreement:

- The minimum number of aircraft retrofitted to obtain a discount on equipment costs ($N_{DISC}$).
- The minimum discount on equipment costs (*DISC*).

Equation (9) represents how to apply these parameters once all the equipment prices, $C_{EQUIPl}$, are known, and the quantity of aircraft to be retrofitted, $Q$:

$$C_{EQUIP} = DISC \cdot (Q/N_{DISC}) \cdot \sum_{l=1}^{E} C_{EQUIP_l} \\ \text{with} (Q/N_{DISC}) \in Z \tag{9}$$

$E$ represents the number of equipment items to be purchased. In this case, $(Q/N_{DISC})$ represents the division without a remainder. The suggested values for $N_{DISC}$ and *DISC* are 50 and 0.05. It means that a discount of 5% is applied by the manufacturer for every 50 aircraft to be retrofitted. Naturally, an upper limit on the percentage cost reduction must be applied. This should not exceed 50%.

### 3.4. Capital Costs

Generally, non-recurring costs are initially apportioned over the first few units produced; therefore, the first aircraft to be upgraded will have a higher price than the rest. With the aim of determining the aircraft operating costs and performing a cost-benefit analysis of a retrofit program, it was decided to distribute the non-recurring cost uniformly among all aircraft of the retrofitted fleet. A gain for the manufacturer has also been considered through a profit margin, shown as a percentage factor. According to Pearce and Brian [34], an aircraft manufacturer generates an average return on invested capital (ROIC) of 7% (as assumed in the application section of the current paper). This can be used as a proper percentage factor. The amount of the total development cost to be allocated to each refurbished aircraft $AC_{DEV}$ has been calculated using Equation (10):

$$AC_{DEV} = \frac{C_{DEV}}{Q}(1 + PRF_{DEV}) . \tag{10}$$

In this equation, $Q$ represents the number of aircraft that compose the fleet to be retrofitted and $PRF_{DEV}$ is the manufacturer profit margin applied to the development costs. $C_{DEV}$ represents the total retrofitting costs in Equation (4). The recurring cost $C_{CONV}$ and $C_{EQUIP}$, computed in Equation (8) and Equation (9), respectively, already refer to a single aircraft to be retrofitted. In consequence, the capital cost required to retrofit a single aircraft belonging to a specific fleet can be computed using Equation (11). With these data, it is possible to compute and compare the total cost per aircraft for an airliner to retrofit its fleet:

$$C_{CAPITAL} = AC_{DEV} + C_{CONV} + C_{EQUIP} . \tag{11}$$

## 4. Case Study: Methodology Applied to a Regional Jet Aircraft

A methodology that is useful for estimating the possible expenses that characterize the three domains into which the retrofitting cost is divided has been presented in this paper. Through this, it is possible to compute the total investment required to undertake a

general aircraft upgrade. The reference platform considered for the analysis is a regional 90-passenger jet aircraft. Its characteristics are summarized in Table 1.

**Table 1.** The reference regional jet aircraft's main characteristics.

| Aircraft | Characteristics |
|---|---|
| Wing Area | 81.40 m$^2$ |
| Wingspan | 27.20 m |
| Design Mission | 1890 nm + 100 nm + 5% reserve |
| Typical Mission | 720 nm |
| Maximum Take-off Weight | 39,500.00 kg |
| OEW | 23,900.00 kg |
| Payload Mass | 9180.00 kg |
| Fuel Mass | 6450.00 kg (design mission) |
| Engine BPR | 5.4 |
| T0 ISA Sea Level (Single-engine) | 78,200 N |
| OBS | Conventional |

Two "retrofitting packages" are considered:

- High BPR-geared turbofan engine installation, leading to improvements in fuel consumption, noise, emissions and maintenance costs.
- OBS architecture electrification, offering more electric and all-electric architectures (MEA/AEA), improving fuel efficiency, maintenance and costs.

Figure 2 shows the baseline aircraft considered for the analysis, highlighting the components to be retrofitted. The new engines, designed as part of the AGILE 4.0 [35] project (2019–2022), have an advanced architecture (as in the Pratt & Whitney PW1000 G series) and a BPR of between 9 and 15. By contrast, the conventional engine is comparable to the General Electric CF34 engine, with a BPR equal to 5.4. The second PIP involved in the retrofit program is the onboard system architecture. The reference aircraft is characterized by a conventional architecture in which the power generation and distribution systems are hydraulic, pneumatic and electric. From this layout, the idea is to move toward a more electric concept. In this manner, the other three OBS architectures are considered.

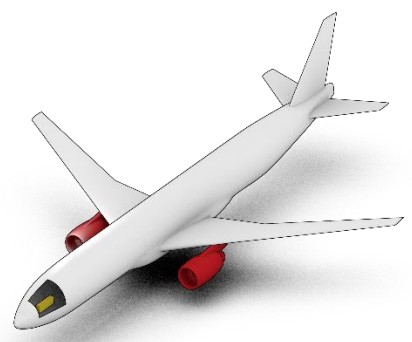

**Figure 2.** Reference example of a regional jet aircraft, with the engines and OBS highlighted.

- More Electric 1 (MEA1). The hydraulic system is completely removed, along with its distribution system. All actuators are electric.
- More Electric 2 (MEA2). The peculiarity of this architecture is represented by the electrification of the wing ice protection system (WIPS) and the environmental control system (ECS). This is a bleedless configuration with electrically driven compressors and hydraulic pumps that are powered by electric motors.
- All-Electric (AEA). An all-electric architecture adopts the innovative features of MEA1 and MEA2; thus, neither the hydraulic nor the pneumatic system are present. No bleed air is required and the pneumatic power is produced by dedicated compressors.

In Table 2, a brief description of the four OBS architectures is presented.

**Table 2.** Main characteristics of the architecture considered for OBS electrification.

| OBS Architecture | Main Characteristics |
|---|---|
| 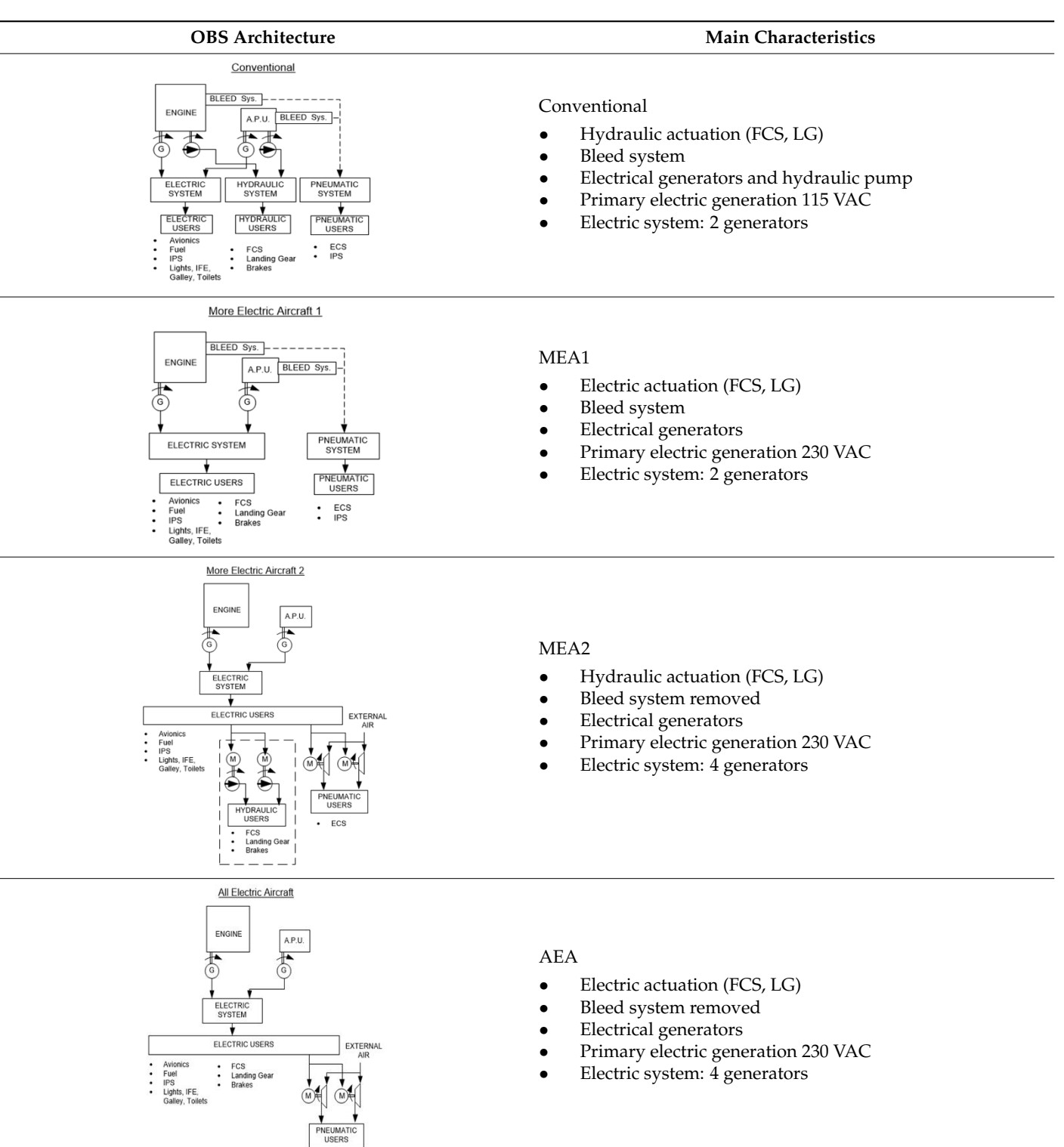 | Conventional<br>• Hydraulic actuation (FCS, LG)<br>• Bleed system<br>• Electrical generators and hydraulic pump<br>• Primary electric generation 115 VAC<br>• Electric system: 2 generators |
| | MEA1<br>• Electric actuation (FCS, LG)<br>• Bleed system<br>• Electrical generators<br>• Primary electric generation 230 VAC<br>• Electric system: 2 generators |
| | MEA2<br>• Hydraulic actuation (FCS, LG)<br>• Bleed system removed<br>• Electrical generators<br>• Primary electric generation 230 VAC<br>• Electric system: 4 generators |
| | AEA<br>• Electric actuation (FCS, LG)<br>• Bleed system removed<br>• Electrical generators<br>• Primary electric generation 230 VAC<br>• Electric system: 4 generators |

In the following paragraphs, a detailed analysis of the retrofitting operations and costs required to install a BPR 15 geared turbofan engine and to upgrade the OBS from a conventional to an AEA configuration is presented. From the results of this analysis, the costs required to undertake a minor level of electrification activity of a retrofitting activity

that considers only one of the two presented packages can be computed. To clarify this aspect, Table 3 summarizes the retrofit solution considered for the estimation analysis of costs. Once the aircraft components to be updated are selected, all the intellectual, practical and economical operations needed to perform the retrofit are evaluated.

**Table 3.** The retrofit solution, considered for cost analysis.

| Retrofit Package | Reference OBS | Electrified OBS | | |
|---|---|---|---|---|
| **Reference Engine** | Baseline | MEA1 | MEA2 | AEA |
| **New Engine** | Engine Upgrade | Engine Up. + MEA1 | Engine Up. + MEA2 | Engine Up. + AEA |

*4.1. Development Activities*

The costs associated with development activities coincide with the non-recurring costs, the initial investment that must be computed to support the retrofit project. Table 4 summarizes the assumptions for computing costs indicated in Table 5. The assumptions have been made via data indicated in Section 3.1, as suggested by the LEONARDO company. In Table 5, a list of the development activities required to perform an engine retrofit and an OBS electrification, divided by categories, is presented. For each operation, an estimation of the number of people and the time necessary to accomplish the task are also indicated. Using these data and Equation (2), it is possible to compute the total development cost related to the aircraft retrofit. Moreover, Figure 3 shows the development cost breakdown, to underline the relative importance of each activity in the case of complete engine and OBS retrofitting. As can be seen, the main expenditures are related to testing (32%), OBS design (29%), traveling, documentation and data management (26%). Around 13% of the development costs are related to flight technologies and structures. It is possible to argue the large amount of data management and the traveling and testing costs (around 58% of the total development costs), but these data are knowledge-based figures from the industrial partners of the AGILE4.0 consortium, meaning that having accurate cost estimations from the beginning of the process is of paramount importance. However, if the engine upgrade activity is discarded, most of the tests considered are then not necessary.

**Table 4.** Data assumed for computing the development costs.

| Development Cost Assumption | |
|---|---|
| Engineering cost | EUR 80/h |
| Flight test cost | EUR 7000/h |
| Wind tunnel test cost | EUR 5000/h |
| RIG test cost | EUR 15 million per test |
| Working time per year | 1760 h |

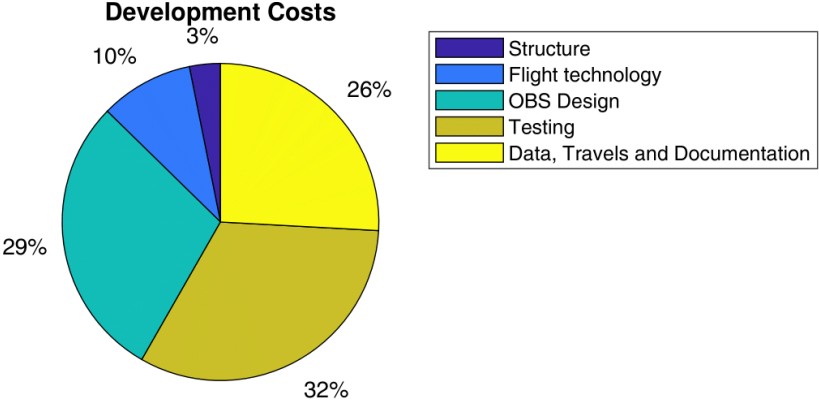

**Figure 3.** Development costs breakdown.

**Table 5.** List of the development activities and their related efforts (in terms of the number of people, time and costs) required to perform the aircraft retrofit.

| Development Activity | | Effort | | |
|---|---|---|---|---|
| **Field** | **Type** | **People** | **Years** | **Costs [Million EUR]** |
| Structure | New engine attachment points | 8 | 1 | 1.1 |
| | Wing stress analysis | 19 | 1.25 | 3.4 |
| | Wing reinforcements design | 9 | 0.75 | 0.95 |
| | Flutter analysis | 3 | 2 | 0.84 |
| | Panels removal and installation | 20 | 1 | 2.8 |
| Flight Technology | Aerodynamics | 10 | 3 | 4.2 |
| | Performance | 15 | 3 | 6.3 |
| | Flight quality | 5 | 3 | 2.1 |
| | Weight and barycenter analysis | 15 | 3 | 6.3 |
| | Structural loads | 20 | 3 | 8.5 |
| OBS Design | Load and failure analysis, new installation drawings | 10 | 3 | 4.2 |
| | Electrical generation/distribution | 18 | 3 | 7.5 |
| | ECS electrical pack | 9 | 3 | 3.8 |
| | Thermal IPS design | 9 | 3 | 3.8 |
| | Air conditioning distribution | 12 | 2 | 3.8 |
| | FCS electrical actuation | 18 | 3 | 7.6 |
| | OBS design, engine installation | 20 | 5 | 14.1 |
| | Engine FADEC, autopilot | 55 | 5 | 38.7 |
| Testing | Wind-tunnel test support | 16 | 1.25 | 2.8 |
| | Flight test support | 20 | 5 | 14.1 |
| | System tests on the complete A/C | 34 | 1 | 4.8 |
| | Ground vibration-resonance test | 4 | 0.2 | 0.11 |
| | Wing static test and support | 38 | 0.75 | 7.5 |
| | Flight test | - | 200 h | 1.4 |
| | Wind tunnel test | - | 500 h | 2.5 |
| | RIG test (4×) | - | - | 60 |
| Documentation | | 15 | 3 | 6.3 |
| Data Management | | 20 | 10 | 28.8 |
| Staffing | | 38 | 5 | 26.8 |
| Travels—Information technology | | - | - | 12.5 |
| **TOTAL** | | | | 287.6 |

In Appendix A, a brief description of the activities shown in Table 5 is presented.

The data illustrated in Figure 3 and Table 5 refer to a retrofitting activity that considers simultaneously both an engine upgrade and the total electrification of the OBS architecture (the AEA architecture is described in Table 2). In this way, the most demanding and expensive activity is considered. To consider more than one retrofitting possibility, other airframe upgrade solutions are also considered. The OBS electrification can be partial, as explained earlier for the MEA1 and MEA2 architectures, or it cannot be included in the retrofitting plan, leading to an isolated engine replacement activity. In the same way, the retrofitting of only the OBS architecture can be performed and analyzed. An overview of the development costs required to perform the possible retrofitting activities described is presented in Table 6 and Figure 4. These data are achieved through a modification of the costs related to the operations introduced in Table 5. According to the retrofitting activity under examination, certain items in the list are removed or are considered less time-consuming and, therefore, less expensive. For instance, if the retrofit consists of the electrification of the OBS, all the engineering and testing activities related to the wing analysis, reinforcements and tests are not considered. Similarly, the incomplete electrification of the systems brings a

reduction in the operations required to analyze, design and test them. For all the retrofitting solutions represented in Table 6 and Figure 4, the activities related to the retrofit design and analysis represent the largest share, between 40 and 50%. The trend is quite similar for testing and other activities, the value of which is around 30 and 20%, respectively. It is possible to see that the most economically convenient airframe upgrade, in terms of development expenses, is represented by the engine replacement alone. Naturally, this will bring reduced benefits in terms of the aircraft's retrofit. Something similar happens in the case of OBS electrification, whereas the retrofit of both the engine and OBS obviously brings an increase in the total cost that is lower than the algebraic sum of both activities. As clearly visible from Table 6 and Figure 4, the design and analysis stage, which represents the conception of a retrofit, has a cost impact of about 40–50% on the total development costs. This amount decreases when increasing the level of retrofitting, due to an increment in the number of rig tests to be performed. The overall support testing phase has a 25–35% impact on development costs. Data, travels, management and documentation represent the last 20–30% of development costs. The bar diagram illustrated in Figure 4 clearly shows the millions of Euros in investment to be overtaken by the OEM, with a range of from EUR 161 million, for only an engine retrofit, up to EUR 288 million for both engine and OBS retrofitting. As stated in Section 1, the impact of the development costs will be spread over the quantity of retrofitted aircraft: the greater the number, the lesser the impact on capital costs.

**Table 6.** List of the main development costs, computed for seven different retrofits.

| Retrofitting Activity | Non-Recurring Development Costs (Million EUR) | | | |
|---|---|---|---|---|
| | Design and Analysis | Testing | Data, Travels and Documentation | Total |
| Engine Upgrade | 77.1 (48%) | 43.4 (27%) | 40.4 (25%) | 160.9 |
| MEA1 | 86.7 (48%) | 55.2 (30%) | 40.4 (22%) | 182.3 |
| MEA2 | 89.9 (53%) | 39.2 (23%) | 40.4(24%) | 169.5 |
| AEA | 103.2 (51%) | 57.6 (29%) | 40.4 (20%) | 201.2 |
| Engine Up. + MEA1 | 103.7 (38%) | 90.8 (34%) | 74.4 (28%) | 268.9 |
| Engine Up. + MEA2 | 106.7 (42%) | 74.8 (29%) | 74.4 (29%) | 255.9 |
| Engine Up. + AEA | 120 (42%) | 93.2 (32%) | 74.4 (26 %) | 287.6 |

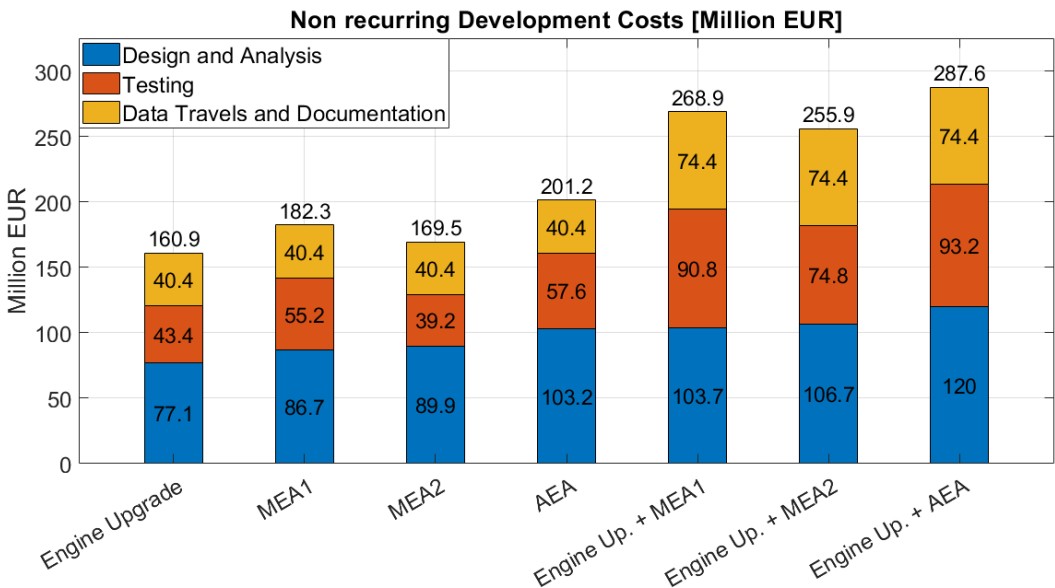

**Figure 4.** The trends of the main development costs required for seven different retrofits.

### 4.2. Conversion Activities

The appropriate modifications and updates to be applied to the fleet are defined and scheduled during the engineering phase. The major conversion operation items involved in the retrofit program under consideration are described in this section, along with details of the labor time and cost required for each operation. In Appendix A, a brief description of the activities presented, divided by categories, is provided.

Starting from the data shown in Table 7, the conversion costs are computed. These are described in Table 8 and in Figure 5. These data are used to compute the total cost of the removal, modification and installation of the upgraded components. For each one of these activities, minor time, manpower and related expenses are considered in the case of partial electrification with and without an engine replacement. In this way, the results illustrated in Table 9 and Figure 6 are obtained. These results show that more than 40% of the total conversion cost is due to OBS replacement and installation. This is due to the heavy impact of this modification at the aircraft level. Around 30% of the conversion costs are due to the engine replacement. Finally, another 30% are due to materials supply, data management and traveling. The bar diagram in Figure 6 highlights how the conversion costs range from EUR 6.6 million for only an engine retrofit up to EUR 14.5 million for an engine plus AEA OBS retrofit. It is worth noting that dismantling the hydraulic system, as in MEA1, leads to higher conversion costs compared to that of an MEA2 OBS architecture at about EUR 0.8 million. This is due to the lower amount of fuselage and floor panel removal required to dismantle the hydraulic systems.

**Table 7.** The data assumed to compute the conversion costs.

| Conversion Cost Data | |
| --- | --- |
| Engine replacement operators | 50 |
| OBS replacement operators | 60 |
| Operation cost | EUR 80/h |
| Working time per month | 160 h |

**Table 8.** List of the conversion activities and their related effort (in terms of the number of people, time and costs) required to perform the aircraft retrofit.

| Conversion Activity | | Effort | |
| --- | --- | --- | --- |
| **Field** | **Type** | **Months** | **Costs (Million EUR)** |
| Engine Removal | Pylon, engine, nacelle | 1 | 0.64 |
| | Wing skin panel | 1 | 0.64 |
| Engine Modification | New engine attachments points | 2 | 1.28 |
| | Spar, ribs, skin reinforcement | 0.5 | 0.32 |
| Engine Installation | Pylon, engine, nacelle | 1 | 0.64 |
| | Wing skin panel | 0.5 | 0.32 |
| OBS Removal | Fuselage skin panel | 1 | 0.77 |
| | Hydraulic distribution | 0.5 | 0.38 |
| | Bleed distribution | 0.5 | 0.38 |
| | Seats, interiors, floors | 2 | 1.54 |
| OBS Installation | Electrical distribution and generation | 2.5 | 1.92 |
| | ECS, IPS, APU, TPs | 1 | 0.77 |
| | Fuselage skin panels | 0.5 | 0.38 |
| Others | Materials | - | 3.0 |
| | Travels | - | 1.0 |
| | Reception, painting and delivery | - | 0.5 |
| **TOTAL** | | | 14.48 |

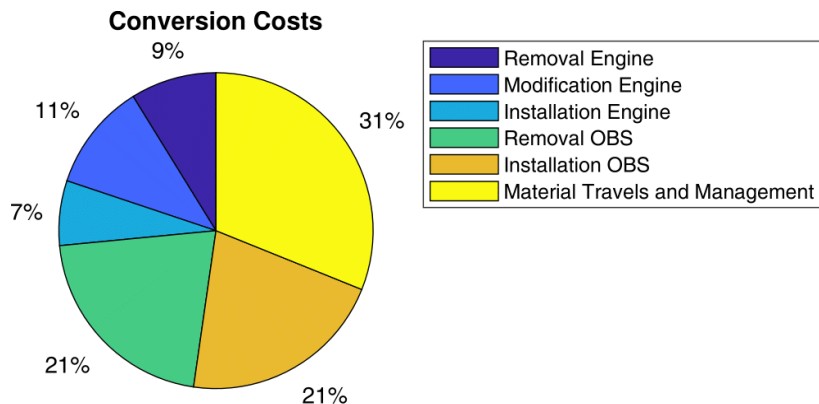

**Figure 5.** Conversion costs breakdown.

**Table 9.** List of the main conversion costs required for every single aircraft to be upgraded, computed for seven different retrofitting activities.

| Retrofitting Activity | Recurring Conversion Costs (Million EUR) | | | |
| --- | --- | --- | --- | --- |
| | Engine Conversion | OBS Conversion | Material Travels and Management | Total |
| Engine Upgrade | 3.84 (61%) | 0.46 (7%) | 2.05 (32%) | 6.35 |
| MEA1 | 0 (0%) | 3.26 (57%) | 2.45 (43%) | 5.71 |
| MEA2 | 0 (0%) | 2.46 (50%) | 2.45 (50%) | 4.91 |
| AEA | 0 (0%) | 5.68 (70%) | 2.45 (30%) | 8.13 |
| Engine Up. + MEA1 | 3.84 (32%) | 3.72 (31%) | 4.50 (37%) | 12.06 |
| Engine Up. + MEA2 | 3.84 (34%) | 2.92 (26%) | 4.50 (40%) | 11.26 |
| Engine Up. + AEA | 3.84 (27%) | 6.14 (42%) | 4.50 (31%) | 14.48 |

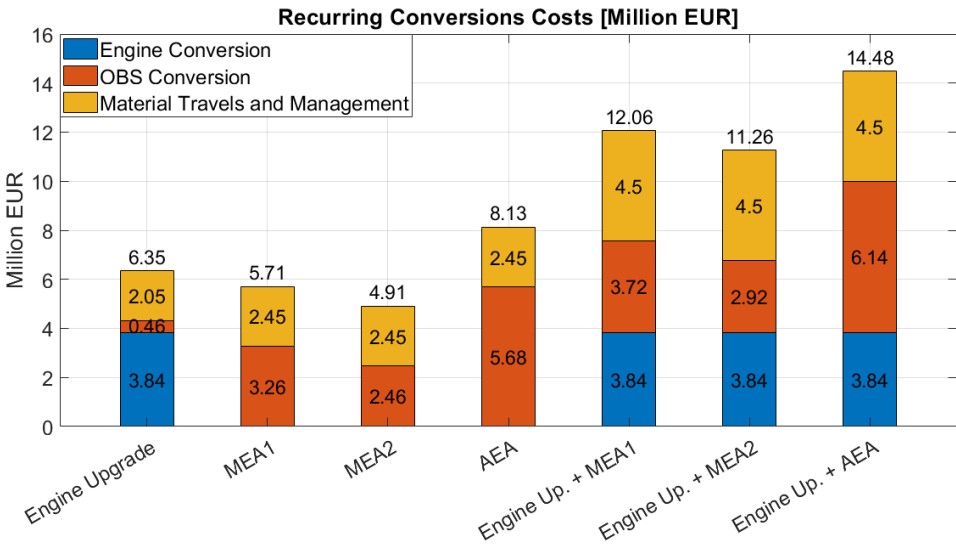

**Figure 6.** The trend of the main conversion costs required for every single aircraft to be upgraded, computed for seven different retrofitting activities.

### 4.3. Retrofitting Equipment

In Table 10, a list of the equipment taken into consideration for all the retrofitting solutions is represented. As already described, two different engine architectures are analyzed: a conventional engine with BP R = 5.4 and advanced geared turbofan engines with a higher BPR (9.0 < BPR < 15). The unit price of this component is obtained using the curves represented in Table 7. The two different curves represented are both obtained via

statistics coming from public data [36–38]. Indeed, two different engine price models better represent the dependency between single-engine costs and static thrust. Figure 7 indicates the two models and the equations used to estimate the conventional and innovative engine list prices. The necessity of generating two curves is caused by the strong difference in innovation and technologies employed for the high BPR-geared turbofan; the points represented in Figure 7 demonstrate that for the same level of static thrust, there is a vertical translation between the low BPR price and high BPR price. The price required to provide a new OBS architecture is computed using Beltramo's [22] methodology. Given, as input, the total aircraft and the system weight breakdown, the methodology can compute the OBS costs. In this way, three different unit prices can be considered for the MEA1, MEA2 and AEA architecture, as indicated in Table 10.

**Table 10.** List of the equipment considered for engine replacements and OBS electrification retrofitting. The unit price of every single item is indicated.

| Equipment | Unit Price (Million EUR) |
|---|---|
| Engine BPR 15 | 9.5 |
| MEA1 OBS | 7.5 |
| MEA2 OBS | 8.4 |
| AEA OBS | 8.3 |

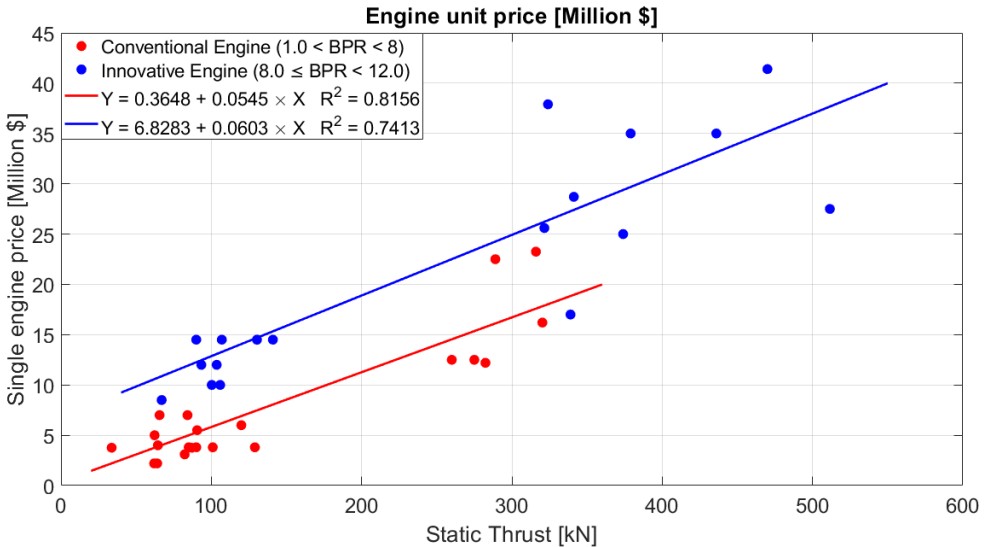

**Figure 7.** Conventional and innovative engine prices and static thrust trend comparison. The points are obtained from available public data.

### 4.4. Regional Jet Aircraft Retrofitting Costs

Table 11 summarizes the total development, conversion and equipment costs required to perform seven different refortifying activities. In case the operation includes an engine upgrade; the turbofan BPR that is considered is 15. Using these data, it is possible to see how each of the three aliquots of the cost increases by considering a more demanding retrofitting activity, which includes both engine and OBS upgrades. The development cost required to perform an engine upgrade represents the lowest value; in contrast, this kind of activity represents a large expense in terms of equipment acquisition, due to the high cost of the innovative turbofans. On the other hand, an OBS upgrade requires a higher cost for developing the retrofit and a lower cost for acquiring the equipment.

**Table 11.** Summary of total development, conversion and equipment costs required for every single aircraft to be upgraded, computed for seven different retrofitting activities.

| Retrofitting Activity | Development Cost (Million EUR) | Conversion Cost (Million EUR) | Equipment Cost (Million EUR) |
|---|---|---|---|
| Engine Upgrade | 160.9 | 6.35 | 19.0 |
| MEA1 | 182.3 | 5.71 | 7.5 |
| MEA2 | 169.5 | 4.91 | 8.4 |
| AEA | 201.2 | 8.13 | 8.3 |
| Engine Up. + MEA1 | 268.9 | 12.06 | 26.5 |
| Engine Up. + MEA2 | 255.9 | 11.26 | 27.4 |
| Engine Up. + AEA | 287.6 | 14.48 | 27.3 |

It is important to underline that the development cost is a non-recurring cost. This means that it must be addressed only once, whatever the number of aircraft to be retrofitted. By contrast, the conversion and equipment costs are recurring costs. They must be calculated for each aircraft upgraded. When an entire fleet must be retrofitted, this cost can be modified by the profit margin, learning curve factor and possible agreements between the supplier and the OEM. In the following section, all these effects will be considered to better estimate the capital costs related to the development of the retrofitting activities and compare it to the savings generated by the aircraft upgrade.

**5. Impact of Retrofitting Activities**

The purpose of this section is to show a cost-benefit analysis, applied in a retrofit case study. Such an evaluation is essential and beneficial during the preliminary design stage to understand the retrofit feasibility. The results from multidisciplinary design analysis and optimization are employed to consider a realistic retrofit process. In particular, the AGILE 4.0 collaborative MDO environment allowed us to highlight the impact of a complex retrofitting activity on a regional jet aircraft. In Table 12, all the hypotheses that were assumed to perform the trade-off analysis are summarized. According to the scenario, the retrofitted aircraft will operate for twelve years or more, during which time it will complete seven flights per day for almost every day of the year. The residual value of the aircraft at the end of its life will be a percentage of the retrofitting costs, mainly due to the new equipment installed that is not yet at the end of its life. The fuel price considered is actualized to the value assumed at the beginning of the year 2022. The noise emissions are computed, considering the current taxes required by Frankfurt airport.

**Table 12.** The hypothesis that is assumed to compute the costs and savings arising from the multidisciplinary retrofitting analysis.

| Costs vs. Savings Analysis Hypothesis | |
|---|---|
| Flights per day | 7 |
| Operative days per year | 358 (a–b check included) |
| Flights per year | 2506 |
| Flight hours per year | 3579 (block time = 1.5 h) |
| Years of utilization | 12 |
| Aircraft residual value | 10% |
| Maintenance saving | (5–10%) |
| Manufacturer profit margin | 7% |
| Learning curve rate | 0.95 |
| Agreement saving | (30–50%) |
| Fuel price | EUR 0.65/kg of kerosene/EUR 88 per barrel |
| Noise taxes | Frankfurt airport taxes |

### 5.1. Parametric Analysis and Results

A multidisciplinary analysis is performed on different retrofitted systems to select the best solution. Both the costs and benefits of an aircraft upgrade activity are considered. Engine and OBS replacement activities are analyzed, either applied together on the same platform or considering the update of just one of them. In addition, four different OBS architectures and four different engines, mainly distinguished according to BPR, are considered. A design of experiment (DOE) is required to study the different architectures' pros and cons easily. Therefore, a multidisciplinary collaborative multi-fidelity workflow system is employed, based on the AGILE 4.0 operational collaborative environment (OCE). The workflow execution is performed via RCE [39,40], using the Brics [39,40] technology to remotely run a disciplinary competence in a collaborative way. The executable workflow, developed in an RCE environment, is shown in Figure 8. According to the type of retrofit under analysis, the engine and OBS architecture are defined and designed during the execution of the workflow, based on surrogate models and semi-empirical methods. In each case, the best engine X and Z positions that minimize fuel consumption are selected. At the end of the workflow, the recurring and non-recurring costs, the aircraft's price and the direct operative costs are computed, employing the method used by Kimoto et al. [22] and AEA [41]. Moreover, the methodology described in Section 3 is implemented to compute the retrofitting's associated costs.

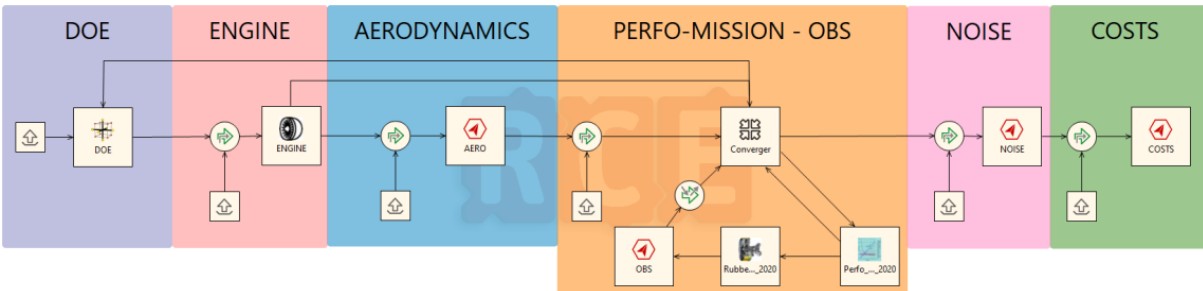

**Figure 8.** Executable AGILE 4.0 workflow in an RCE environment, as an example of a converged DOE.

After the execution of the DOE workflow, the first clear result is represented by a reduction in aircraft fuel consumption. This trend can be seen in Figure 9. The fuel mass required for a typical aircraft mission decreases with the level of OBS electrification. This is mostly due to the strong reduction in the secondary power required by the engine and to the lower weight of the new OBS architecture. On the other hand, the fuel mass consumed during the flight does not necessarily decrease with an increase in engine BPR. However, for a high value of BPR, it remains quite constant. This is mainly due to the overall dimensions and weight of the engine, which increase in a non-negligible way for a high value of BPR. However, the adoption of an advanced engine (with a higher BPR) leads to a fuel reduction ranging from 12 to 14%, while the OBS electrification could lead to further fuel reductions, ranging from 2% for the MEA1 architecture to 4% for the AEA architecture. It is possible to visualize how the overall aircraft weight and emissions are directly influenced by the consumption of fuel, leading to a significant reduction in $CO_2$ and NOx generation. The effect of these advanced engines is also reflected in terms of the noise emitted during the take-off and landing cycles. The results show a cumulative certification noise reduction of about 15 in the effective perceived noise in decibels (EPNdB) for the higher value of BPR considered, which manifests mainly during the take-off phase. The methodology adopted for noise computation follows the ICAO environmental protection standards, in Volume 1 of Annex 16 [42], and part 36 of the noise certification requirements of the Federal Regulations [43]. A more detailed description has been provided by the authors of [44].

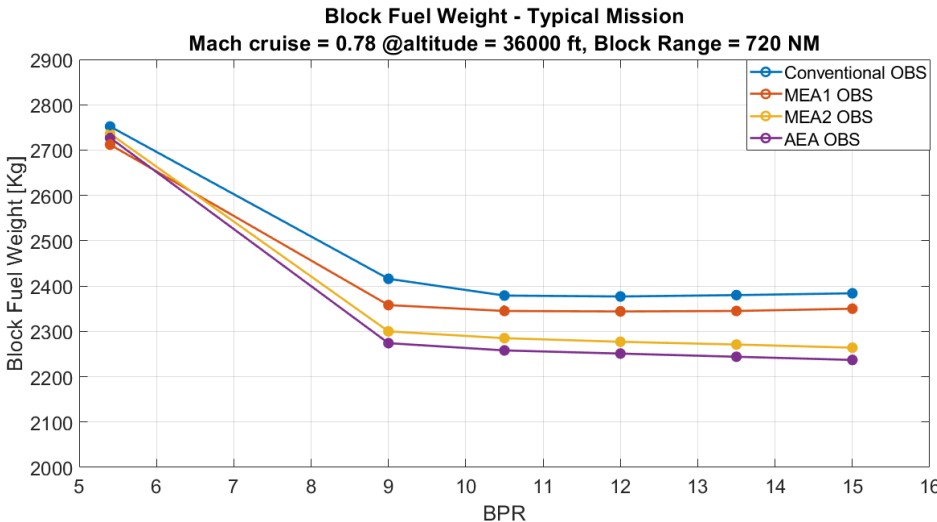

**Figure 9.** The trend of block fuel consumed during a typical mission by a different system solution obtained through the execution of DOE.

Other than their performance, capital costs relating to the development of all the retrofitting activities that are considered in the DOE are computed. Table 13 shows the values computed for a single year of aircraft operation for three different fleets. The system under analysis is the same as that considered in the previous section, a retrofitted aircraft with a BPR 15 engine and AEA OBS architecture. The development costs are distributed among all the aircraft that comprise the fleet. The conversion costs are affected by a learning curve factor that depends on the fleet size. The equipment costs are computed while simulating an agreement between the supplier and the OEM; the related discount is dependent on the number of items acquired, up to a maximum value of 50%. For these reasons, the capital costs are decreased by increasing the number of aircraft retrofitted. The reduction that is reached regarding a fleet composed of 300 aircraft compared to one composed of 700 aircraft is around 20%. In each case, the most convenient choice in terms of costs will be the retrofit of the OBS. In Figure 10, a synthesis of the capital costs evaluated for all the retrofitting activities considered in the DOE is represented. The trend of these costs with reference to the fuel consumed during a typical mission is illustrated. As can be seen, the higher the investment in the retrofitting activity, the higher the reduction in fuel consumed during the flight. A similar trend is followed by the values for CO, NOX and noise emission of the aircraft.

**Table 13.** The capital costs derived from three different retrofitting activities. The costs are expressed in million EUR and are related to a single aircraft belonging to one of three fleets.

| Fleet | Retrofit Type | Profit Margin | Develop. | Conv. | Equip. | CAPITAL |
|---|---|---|---|---|---|---|
| | Engine Upgrade | 0.04 | 0.54 | 4.67 | 10.64 | 14.29 |
| 300 | AEA | 0.05 | 0.67 | 5.65 | 5.81 | 10.97 |
| | Engine Up. + AEA | 0.07 | 0.96 | 10.25 | 16.45 | 24.96 |
| | Engine Upgrade | 0.02 | 0.32 | 4.49 | 7.60 | 11.20 |
| 500 | AEA | 0.02 | 0.40 | 5.45 | 4.15 | 9.03 |
| | Engine Up. + AEA | 0.04 | 0.58 | 9.87 | 11.75 | 20.02 |
| | Engine Upgrade | 0.02 | 0.23 | 4.39 | 7.60 | 11.01 |
| 700 | AEA | 0.02 | 0.29 | 5.32 | 4.15 | 8.80 |
| | Engine Up. + AEA | 0.03 | 0.41 | 9.63 | 11.75 | 19.64 |

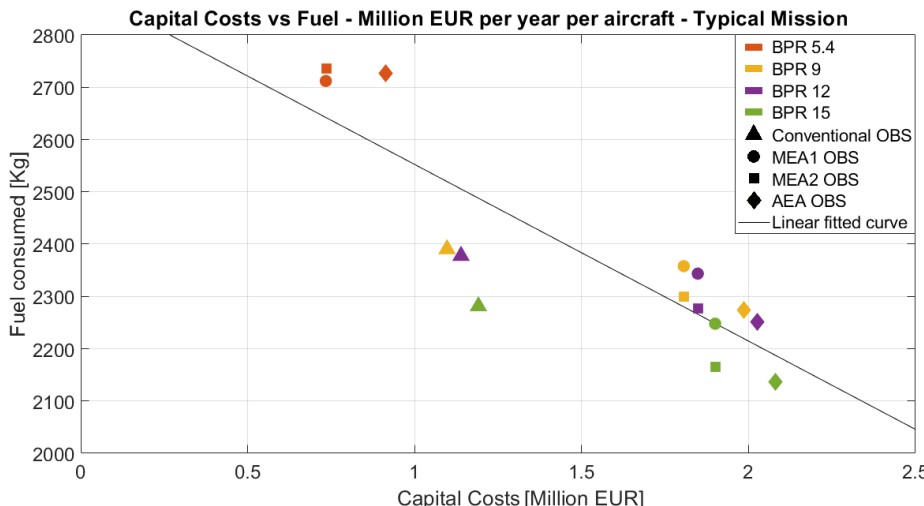

**Figure 10.** Fuel consumed during a typical mission and the capital costs represented by all the retrofitting solutions considered. The color of each point represents the engine BPR, while the shape (triangle, circle, square or diamond) represents the OBS levels of electrification.

In general, an airliner will accept a trade-off choice between a reduction in aircraft emissions and the required investment. To analyze which choice is most convenient, an evaluation of the savings resulting from the retrofitting activities is conducted in the following paragraph.

*5.2. Retrofitting Savings*

Based on our results and the data previously illustrated, it is possible to compute and compare all the savings generated by a particular retrofitting activity. Table 14 shows the total amount in million EUR saved in a single year per aircraft as a consequence of the three aircraft retrofitting activities performed on a given fleet. In this case, different fleet sizes are not computed since the number of aircraft retrofitted does not influence the savings made in a year.

**Table 14.** The savings derived from three different retrofitting activities. The costs are expressed in million EUR per year and are related to a single aircraft belonging to a fleet composed of 300 units.

| Retrofit Type | Engine Upgrade | AEA | Engine Up. + AEA |
|---|---|---|---|
| Δ Fuel (Million EUR) | 0.80 | 0.04 | 1.26 |
| Δ Emissions Charges (Million EUR) | $18.5 \times 10^{-4}$ | $3.9 \times 10^{-4}$ | $3.0 \times 10^{-4}$ |
| Δ Noise Charges (Million EUR) | 0.14 | $1.4 \times 10^{-4}$ | 0.14 |
| Δ Maintenance (Million EUR) | 0.12 | 0.19 | 0.25 |
| SAVINGS (Million EUR) | 1.06 | 0.23 | 1.65 |

All the savings computed in this table are considered as the difference between the costs required to operate the baseline and the upgraded aircraft. The fuel costs are computed, considering the decrease in the amount of kerosene consumed during a typical mission; a price of EUR 88 per barrel has been assumed [45]. The costs related to $NO_X$ and carbon monoxide (CO) emissions are computed in accordance with ICAO Annex 16, Volume II [46]. Conversely, costs due to noise emissions are computed, considering the current taxes required by Frankfurt airport [47]. Maintenance savings are computed according to the data provided by Embraer with respect to the E2 Family costs [5]. Table 12 shows all the hypotheses that are assumed to compute the presented calculation. The system under analysis is the same as that considered in the previous section, a retrofitted aircraft with a BPR 15 engine and AEA OBS architecture. As can be seen, the most convenient

option, in terms of savings, is represented by the engine and OBS replacement. Indeed, a wider-ranging retrofitting activity led to a greater improvement in fuel consumption, emissions and maintenance. In contrast, the electrification of the OBS architecture only brings a significant reduction from a maintenance point of view. This is mainly due to the reduced improvement in fuel consumption and emissions performance.

Table 15 illustrates the difference between capital costs and savings per year per aircraft, obtained for three different fleets and retrofit solutions. The data for the BPR 15 and AEA architecture are also shown. The values indicated in this table are the difference between capital costs, illustrated in Table 13, and savings, shown in Table 14. As it is possible to imagine, the solution is more convenient if the number of retrofitted aircraft is increased. The best solution in terms of the difference between costs and savings is represented by an engine substitution, since the advantages achieved in terms of fuel consumption and emissions become relevant without the need for a high level of investment. Figure 11 shows the data described, via a bar chart. As can be seen, the difference between capital costs and savings is negative for the three different cases considered. Indeed, the engine upgrade activity becomes remunerative only if a fleet composed of more than 300 retrofitted aircraft is considered. Similarly, the savings generated by engine and OBS upgrades only achieve the costs required to perform the activity when considering a 700-aircraft fleet. This is mainly due to the high level of investment required to perform the retrofitting activities analyzed. Therefore, considering the current scenario, the retrofitting activities seem to be economically convenient if an engine upgrade is included and if a high number of aircraft involved in the operations is considered. For the remaining cases, if the authorities impose an abatement of pollutant emissions, the positive difference shown in Figure 11 represents the cost that must be accepted to meet the regulations Figure 12–14 represent three different scenarios, in which the savings due to the innovations introduced into the system overtake the initial retrofitting investment, also when considering a fleet composed of 300 units. In Figure 12, an increased price of fuel up to EUR 120 per barrel (+36%) is considered. Instead, in Figure 13, a +75% increase in the number of EUR per movement that are required by the baseline aircraft to take off or land in Frankfurt airport is considered. In Figure 14, the fuel price is actualized to that in May 2022, a period during which it experienced enormous growth, up to EUR 159 per barrel (+81%). In all these scenarios, the savings due to reduced fuel consumption and emissions are heavily emphasized. This is the reason why the difference between costs and savings reached EUR −0.4 million per year per aircraft in the first two scenarios and EUR −0.8 million per year per aircraft in the last scenario. This happens in both the engine upgrade and engine + OBS upgrade scenarios, but only if a larger fleet is considered. For these three scenarios, the engine upgrade always represents a remunerative activity; in contrast, the OBS upgrade represents a non-economically viable scenario, mainly due to the minimal savings generated by this operation. In the previous scenario, the engine and OBS upgrade becomes more worthwhile than the engine retrofit. This is due to the high fuel price, which emphasizes+ the benefits achieved by a solution that generates the highest value of fuel consumption reduction. Figure 15 summarizes the savings and the capital costs for all the configurations considered via the DOE, assuming a fleet composed of 300 aircraft. From this figure, it is possible to see how the savings derived from a retrofitting activity increase with the investment required to perform the retrofitting, because of the increasing complexity of the aircraft upgrade.

It is worth noting that similar considerations are valid not only for regional but also for short–medium-range aircraft. In this respect, an assumption of a 700-aircraft fleet appears realistic, opening up the possibility of further increasing the number of potential retrofitted aircraft.

**Table 15.** Difference between capital costs and savings, related to three different retrofit solutions and three different fleet sizes.

| | | Capital−Savings (Million EUR Per Year) | | |
|---|---|---|---|---|
| | | Number of Aircraft | | |
| | | 300 | 500 | 700 |
| | Engine Upgrade | 0.13 | −0.13 | −0.14 |
| Retrofit Type | AEA | 0.68 | 0.52 | 0.50 |
| | Engine Up. + AEA | 0.43 | 0.02 | −0.013 |

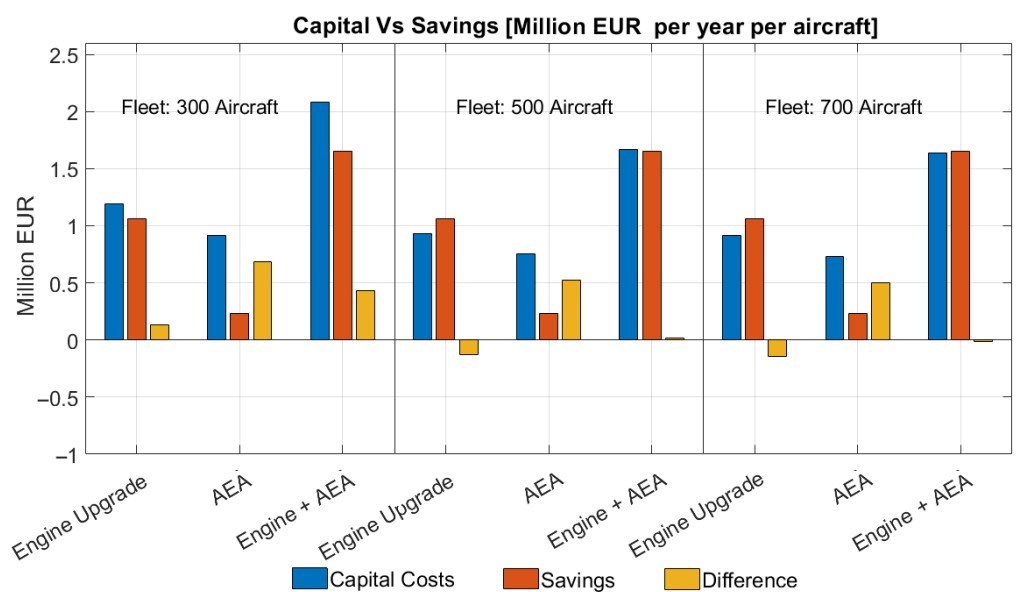

**Figure 11.** Capital costs and savings and their differences per year per aircraft. A positive difference means that capital cost overcomes savings. Data are shown for three different fleets and three different retrofit solutions.

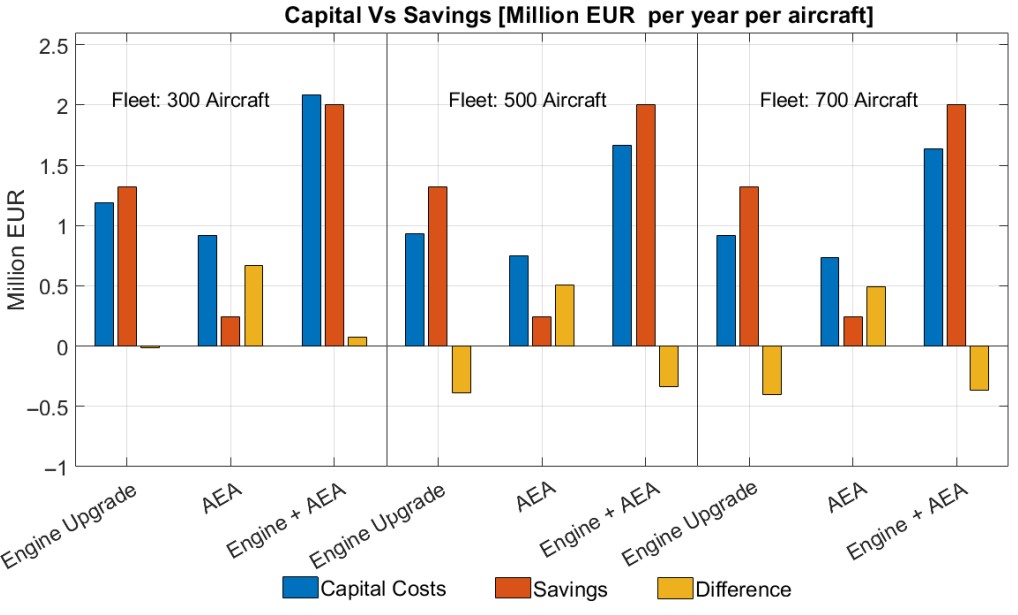

**Figure 12.** Capital costs and savings and their difference per year per aircraft. The fuel price considered is EUR 108 per barrel, +36%, as taken from Figure 11.

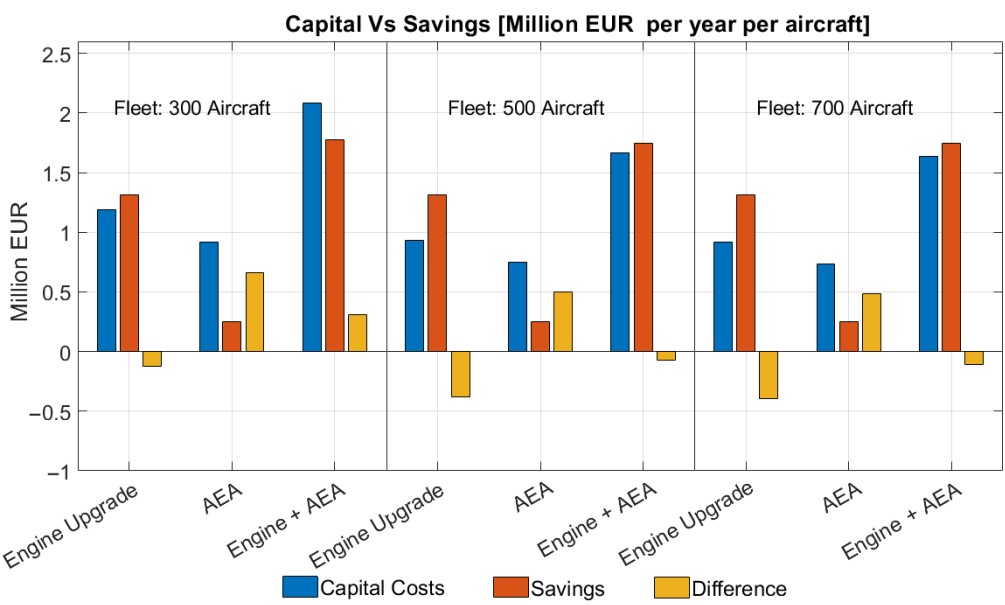

**Figure 13.** Capital costs and savings and their difference per year per aircraft. The noise taxes considered for the baseline are increased by 75%, as taken from Figure 11.

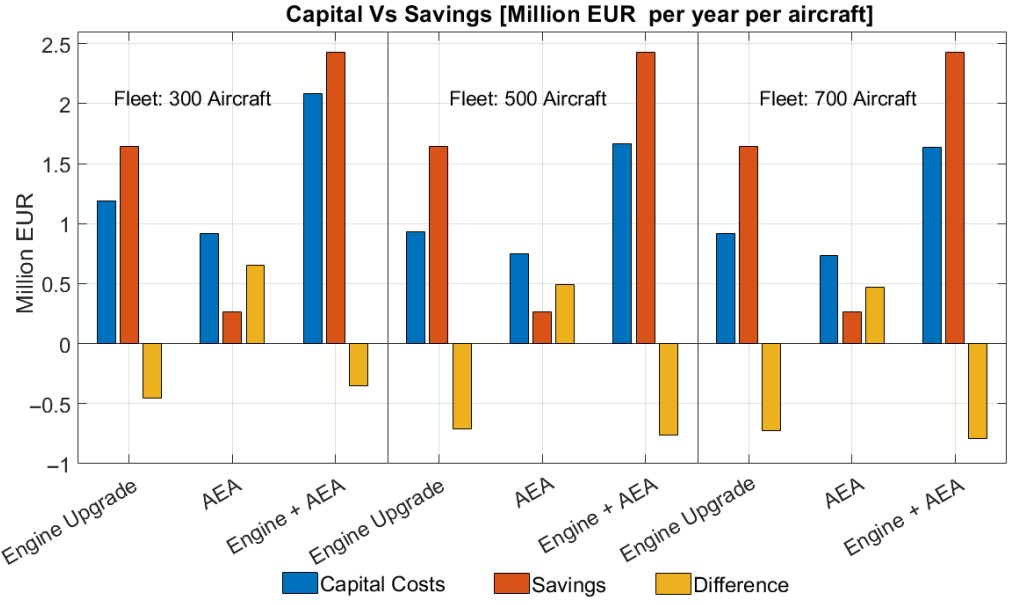

**Figure 14.** Capital costs and savings and the difference per year per aircraft. The fuel price considered is EUR 159 per barrel, +81%, as taken from Figure 11.

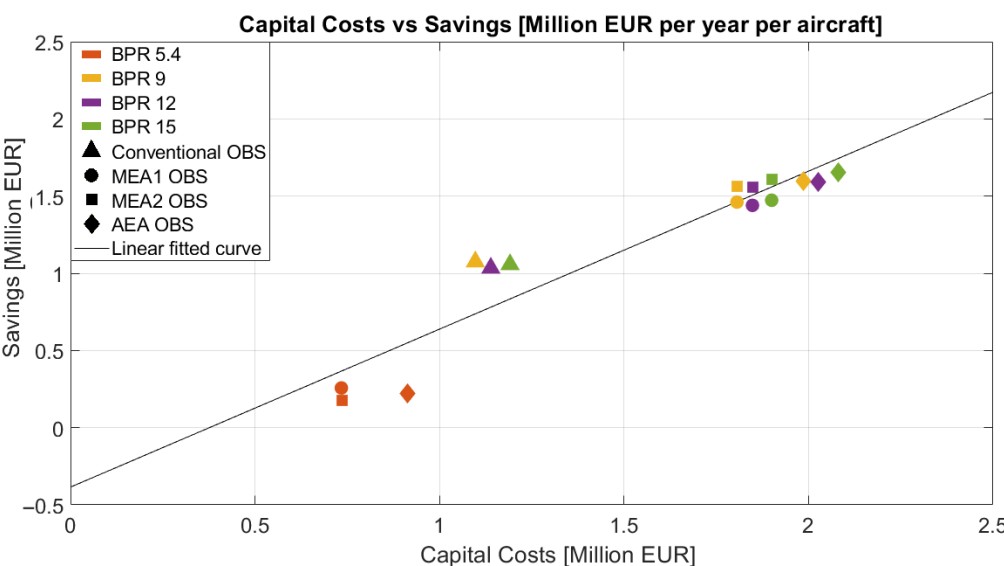

**Figure 15.** Capital costs and savings, represented for all the retrofitting solutions considered. The color of each point represents the engine BPR, while the shape (triangle, circle, square or diamond) represents the OBS level of electrification. The assumptions made to obtain the data are indicated in Table 12. A fleet composed of 300 aircraft is assumed.

## 6. Conclusions

A methodology that is useful for estimating retrofitting costs at an industrial level has been provided, illustrating all the activities related to such a process. Costs are mainly derived from three contributions: development costs, conversion costs and equipment acquisition costs. Through this approach, the impact of an engine and OBS replacement and modernization process on a regional jet platform has been presented. The AGILE 4.0 architectural framework gave the possibility of performing a DOE analysis that was able to compute the effect on the performance and costs of the novel designed platform. From the performance and savings point of view, all the retrofit activities considered are desirable. Significant reductions in fuel consumption, emission, noise and the costs generated during aircraft operations are obtained. Of course, the costs required to accomplish the retrofit increase with its complexity and, consequently, with the improvement reached. Therefore, for all the retrofitting activities reported, the expenses saved due to the upgrade only overtake the investment required to carry on all the activities if a large fleet of retrofitted aircraft is considered. It has been shown that in a scenario in which the price of the fuel or the emissions taxes are increased, the retrofitting activity becomes economically viable for a smaller fleet. For example, a break-even point for engine upgrade activity can be reached if the fuel price increases from EUR 88 per barrel (the actualized value) to EUR 95 per barrel (a level already reached in 2014). Moreover, if the fuel price rockets up to EUR 159 per barrel, as happened in May 2022, the retrofitting activity becomes highly viable from both an economic and performance perspective. However, there is the concrete possibility that the authorities will impose rules for an abatement of pollutant emissions in the near future. The engine upgrade activities considered, both together and without onboard system electrification, represent a solution that allows engineers to significantly improve the air and noise emissions. In the above-described scenario, they also represent some examples of economically remunerative activities. In contrast, the onboard systems electrification activities represent a technologically viable solution that may not be economically realistic. Indeed, the emission benefits of such an operation are not very high in terms of the investment required to undertake the aircraft upgrade. In addition, in all the scenarios considered, onboard systems electrification is always not economically viable for the airline, which may choose to reject the upgrade activity. The main limitation of the methodology presented concerns the data required to perform the estimations. Indeed,

the retrofitting cost data provided in this paper come from industry knowledge and are not always easy to source. In addition, coefficients such as the operation's hourly cost or the test cost are strongly dependent on the context in which the activity is performed. As a consequence, their choice must be appropriate in terms of the scenario under analysis. The outcome of the present work could address a preliminary estimation of the effort that airlines and manufacturers will face in complying with the regulatory limitations.

**Author Contributions:** Conceptualization, P.D.V.; methodology, M.M. and P.D.V.; software, M.M.; validation, M.M. and P.D.V.; formal analysis, M.M.; resources, M.M., P.D.V. and V.C.; data curation, M.M., P.D.V. and V.C.; writing—original draft preparation, M.M., P.D.V. and V.C.; writing—review and editing, M.M., P.D.V. and V.C.; supervision, F.N.; project administration, F.N.; funding acquisition, F.N. All authors have read and agreed to the published version of the manuscript.

**Funding:** The research presented in this paper has been performed within the framework of the AGILE 4.0 project (Toward Cyber-physical Collaborative Aircraft Development) and has received funding from the European Union Horizon 2020 Program under grant agreement number 815122.

**Conflicts of Interest:** The authors declare no conflict of interest.

**Appendix A**

In this section, a brief description of the development activities shown in Table 5 is presented.

Structures. Before the installation of the new technology on an existing aircraft, the operations must be supported by engineering efforts focused on the modification of the airframe structure. All the components that are set to remain unaltered do not need any structural modifications (i.e., reinforcements/redesign). The studies deal with the following aspects:

- New engine attachment points. New engines may be installed on different wing attachment points compared to the previous ones. A higher bypass ratio means that the fan size is increased; as a result, mounting these engines under a wing could be a challenging task that requires great engineering effort.
- Wing stress analysis. Due to the different geometries and characteristics of the new engines, the inertia, force and thrust generated will certainly change. The static aeroelastic deformation of the wing structure and load distributions, bending moment and torque need to be studied. For this purpose, a new structural finite-element model of the wing/engine system must be established.
- Wing reinforcement design. A possible conclusion of the wing stress analysis may be the realization that a wing reinforcement is needed, due to the issues described in the previous points.
- Flutter analysis. The engine module position modification along the wing in both spanwise and chordwise directions can influence the flutter characteristics. The natural vibration modes of the structure may also change with the adoption of new actuators. The structure should be capable of supporting this at the critical loads present on the maneuvering diagram.
- Panel removal and installation. The hydraulic and pneumatic circuits run across the wings and the fuselage, to connect the energy sources to the various users. If the onboard systems are modified, it is necessary to remove the fuselage panels and reinstall them after the replacement. The engineering effort will be focused on planning the operations of the fuselage panel disassembly and assembly.

Flight Technology.

- Aerodynamics. A computational fluid dynamics (CFD) analysis must be carried out to predict the drag, lift, noise, performance, structural and thermal loads for the updated aircraft systems.
- Performance. The aircraft mass distribution is an important parameter to be considered during the design process, due to its significant influence on performance and inertia.

If the new engines are located at a greater distance from the fuselage, they will make a greater contribution to the rolling moment of inertia of the aircraft. In addition, their weight and efficiency changes will all influence the aircraft's mass distribution.

- Flight quality. A certain amount of engineering effort is involved in the study and in the evaluation of the longitudinal and lateral-directional stability and control characteristics of the retrofitted aircraft.
- Weight and center-of-gravity analysis. The proper distribution of weight plays a large and important role in an aircraft's overall performance. Both performance and stability depend on the location of the center of gravity. Therefore, all flight tests must be conducted with an accurate knowledge of the location of the center of gravity at any one point in time.
- Structural loads. An analysis that is performed on all the aircraft in terms of the new structural loads is required for certification purposes and to understand if reinforcing element installation is required.

OBS Design. The partial or total electrification of the onboard systems requires intensive engineering work, aimed at designing the new architecture.

- Electrical generation/distribution. Power must be provided by an additional electrical generator and distribution system. These components must be sized appropriately and relocated along the aircraft.
- ECS, IPS, air conditioning, FCS. All the components that connect to the new electrified system must be redesigned.
- Load and failure analysis and new installation drawings. For the overall OBS architecture installation, failure analysis must be performed, and new component drawings must be provided.
- OBS design, engine installation, engine FADEC, and autopilot. The simultaneous engine and OBS upgrades imply the installation of a new FADEC (full-authority digital engine control) system and new autopilot software.

Testing. Certification of the new system is essential after the execution of the retrofitting modifications. All the activities required to set up, assist and analyze the tests and test results must be accounted for. Of course, in the case of reduced retrofitting activities, certain test operations that were considered become unnecessary.

- Wind tunnel tests. Once the new engine has been chosen, the combination airframe and new engine must be tested. A wind-tunnel test campaign must be organized and carried out to predict the aerodynamic performance of individual aircraft components, as well as the new overall configuration. The engineering effort required to process the test data and to obtain the new drag polar curves is also considered.
- Flight tests. After the retrofit updates, a flight test campaign is carried out to determine the new aircraft characteristics (previously estimated via wind tunnel tests), to assist the engineering design and developmental process and to verify the attainment of technical performance specifications and objectives, to establish the system's operational effectiveness and operational suitability.
- System tests on a complete A/C and RIG test. Several test systems must be assessed to analyze the behavior of the new onboard systems, starting from the standalone component up to its integration into the aircraft. Four different RIG tests must be performed: tests of the electrical, propulsion, avionic and flight control systems. In addition, an avionics software development process is required by law. This cost item must be considered since the engine's FADEC and the autopilot are changed.
- Ground vibration-resonance test. The modifications to the structure and mass distribution could bring the necessity of new ground vibration tests, performed to meet certification requirements.
  Technical documentation. After such an innovation, it is essential to make an engineering effort to update the various aircraft manuals: the repair manual, the aircraft flight

manual (AFM), the flight crew operating manual (FCOM), and the weight and balance manual (WBM).

Data management. This cost item includes the engineering effort required to control the configuration and manage the data by people who handle information such as onboard equipment serial numbers and the way that these systems interface with the structure. Typically, this activity lasts almost all the life of the upgraded aircraft, which is the reason why this cost can be elevated.

Staffing. In this value are included all the people who have not yet been considered: airworthiness, reliability, maintainability and testability engineers, safety and chief engineers, and the people who deal with design quality assurance, costs and planning.

Traveling and information technology. In this cost item are allocated the materials to support engineering research and the costs to sustain every kind of travel (e.g., the movements of goods and supplies). Travel costs are calculated, as established by Equation (4). The cost associated with information technology is linked to the number of licenses required. A realistic value may be EUR 20,000 per license.

In the following paragraphs, a brief description of the conversion activities presented in Table 8, divided into categories, is presented.

Removal. The entire propulsion system must be removed, including the components attached to the wing. According to the retrofit typology, the hydraulic or the bleed distribution systems must be dismounted. To allow these activities, a preliminary operation that consists of panel disassembly is required. These must be removed from the wing and all along the fuselage, including the floor, beneath which some of the cables and systems components are located. As a consequence, the fuselage interiors must also be removed.

Modification. After the engineering studies are complete, there is a very high chance that the wing must be reinforced. Indeed, the presence of a new engine that can be moved to a different position may lead to new torsional and gyroscopic loads. Furthermore, new engine attachment points must be created since the new engine will have different dimensions from the previous one.

Installation. The new engine and the new electric system must be inserted into the aircraft, including new power generators. The components to which the new system is linked must also be modified, in order to ensure OBS compatibility. Finally, the panels that were previously removed from the wing and the fuselage must be reinstalled.

Material, travel and management. The different and complementary activities required to support the removal, modification and installation phases must be considered. To compute this cost, a formula such as Equation (4) has been utilized by modifying its characteristic factor. A value of EUR 24/h per worker has been considered for the computation of material costs. The value that is assumed to compute travel costs is EUR 8.0/h per worker. Finally, the parameter used for other activities, such as the reception, painting and delivery of the new aircraft is EUR 4.0/h.

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
