# Peer review of "Retrofitting Cost Modeling in Aircraft Design"

_aerospace, doi:10.3390/aerospace9070349_

Round 1

Reviewer 1 Report

Dear Authors,

congratulation to the nice work performed and the extensive work described in the paper!

While you address in your cost-savings analyses uncertainty factors by e.g. fuel prices and airport taxes, I (as an expert within aircraft design but no special expertise in cost predictions) miss an equal analyses (or at least a discussion) about uncertainties and thrust worthiness of your "cost input" data related to work effort etc. regarding the retrofitting scenarios presented in Table 2. While engine retrofitting can be seen as a SOTA operation, from my point of view it is very questionable whether it is at all possible to retrofit an existing SOTA onboard system into a MEA(1 or 2) or even an AEA. Can there really all components installed in the existing airframe? Is it possible to comply with all separation rules etc.?! I would like to see such "risk" assessment in the cost-benefit calculations as well. Maybe, as an simple change, you add a "risk and credibility" section in your Conclusion/Discussion?!

You use the abbreviation "Mln" (and sometimes MLN) for "million" (mega). Please consider to use the standard SI-Unit "M" (e.g. M$) instead! (e.g. in Table 4 and throughout the whole paper)

Also, I identified the following (mainly spelling & formatting) flaws:

Line 34: Cabon Di-Oxide --> Dioxide

Line54: listing aircraft types --> Use order Airbus, Boeing & Embraer! (as listed in the sentence before)

Line119: consider writing "2,000" and "6,000" instead of "2 thousand" and "6 thousand"

Line139 and on several occasions: You have on some locations a "." behind you reference event thought the sentence continues after the reference!

Table1: Extreme high-accurate data for a concept airplane!!! I would (at least fore the weights) remove the after-comma values (if not even more, e.g. to rounding to 10ths of kg, so instead of OEW 23671.15 kg --> 23,370 kg).

Figure7: Would be nice to state the origin of the data points ("obtained from available public data")

Author Response

Dear Authors,

congratulation to the nice work performed and the extensive work described in the paper!

While you address in your cost-savings analyses uncertainty factors by e.g. fuel prices and airport taxes, I (as an expert within aircraft design but no special expertise in cost predictions) miss an equal analyses (or at least a discussion) about uncertainties and thrust worthiness of your "cost input" data related to work effort etc. regarding the retrofitting scenarios presented in Table 2.

In Table 2 are reported the main characteristics of the architectures considered for OBS electrification. The costs related to the retrofitting of OBS architecture have been carefully evaluated at industrial level together with LEONARDO aircraft partner of AGILE4.0 project. Differences have been accounted for all OBS architectures in terms of costs, mainly highlighting the overall needed operations to retrofit the systems. Based on the industrial partner experience (i.e., LEONARDO) in the paper the OBS costs for the four different architectures have not been parametrized, due also to the difficulties to assume reliable costs associated to them. In other words, mainly for the OBS costs, we trusted in the overall LEONARDO data and assumptions (Presented in Table 4, Table 5, Table 6, Table 7, Table 8 and Table 9 and detailed in the Appendix).   

While engine retrofitting can be seen as a SOTA operation, from my point of view it is very questionable whether it is at all possible to retrofit an existing SOTA onboard system into a MEA(1 or 2) or even an AEA. Can there really all components installed in the existing airframe? Is it possible to comply with all separation rules etc.?! I would like to see such "risk" assessment in the cost-benefit calculations as well. Maybe, as an simple change, you add a "risk and credibility" section in your Conclusion/Discussion?!

Yes, you are right. As you can see also in the main conclusion and discussion, the retrofitting of OBS alone is completely economically non convenient. As highlighted in the appendix A, all the engineering, equipment and operational activities have been carefully evaluated (thanks to the help of LEONARDO company) in the cost’s evaluation. From Original Equipment Manufacture (OEM) prospective the OBS retrofitting is technically feasible but again is economically not convenient as also demonstrated in part of the conclusion. However, the OBS electrification is a viable process to reduce the environmental footprint, especially on the new aircraft serial numbers. Usually, the OEM approaches this kind of activities on existing aircraft considering only part of the OBS (i.e., avionics, cabin, hydraulic/electric actuation). Then, based on the experience gained, the electrification of the overall OBS architecture can be really exploited on a new aircraft design.

In the conclusion, a risk and credibility" section has been added to better clarify what is written above.

You use the abbreviation "Mln" (and sometimes MLN) for "million" (mega). Please consider to use the standard SI-Unit "M" (e.g. M$) instead! (e.g. in Table 4 and throughout the whole paper)

Done. In text, tables and figures.

Also, I identified the following (mainly spelling & formatting) flaws:

Line 34: Cabon Di-Oxide --> Dioxide

Modified.

Line54: listing aircraft types --> Use order Airbus, Boeing & Embraer! (as listed in the sentence before)

Modified.

Line119: consider writing "2,000" and "6,000" instead of "2 thousand" and "6 thousand"

Modified.

Line139 and on several occasions: You have on some locations a "." behind you reference event thought the sentence continues after the reference!

Modified. Also in line 218.

Table1: Extreme high-accurate data for a concept airplane!!! I would (at least fore the weights) remove the after-comma values (if not even more, e.g. to rounding to 10ths of kg, so instead of OEW 23671.15 kg --> 23,370 kg).

Modified.

Figure7: Would be nice to state the origin of the data points ("obtained from available public data")

We added some references in line 635 to specify the sources of data points.

Reviewer 2 Report

The authors model the costs required for an engine retrofit on an aircraft considereing the fleet. Overal interisting but the article is very long. Please consider shortening the text by writing the messages more concise.

The manuscript is only about the engine. Thus, this should reflect in the title.

The authors should add a paragraph to the second chapter that retrofitting of an entire engine is not the only way how to reduce the noise emission. There are plenty of other options, like adding chevrons (Rask et al. (2011) "Understanding how chevrons modify noise in supersonic jet with flight effects") or fluidic injection (Semlitsch et al. (2019) "Transforming the shock pattern of supersonic jets using fluidic injection") having the advantage to be only appied during take-off and approach.

Please check your equations. In Eq. 5, the authors write Np_k and Oh_k but they are described as N_pk and O_hk. See also line 407.

There are some details that feel questionable. Like how do you want to mount a BPR 15 engine on a regional jet?

There are some minor language errors. For example, "to show a costs-benefits analysis" -> "to show a cost-benefit analysis".

In line 213: "However, the engine price is not a factor that really influences the engine choose, ..." -> choice

In line 216: there is an additional dot

In line 239, "Beltramo et alii [20]" there some "i" too much. Same errors can be found several times in the manuscript: "The reports developed by Beltramo et alii [20] and Large 385 at alii [23] ...". I guess Large 385 is not a name ... .

In lin 723: Why is Effective captitalised?

 '... for seven different retrofittingS' or '... for seven different retrofits'

Author Response

The authors model the costs required for an engine retrofit on an aircraft considereing the fleet. Overal interisting but the article is very long. Please consider shortening the text by writing the messages more concise.  

Thanks for the suggestion. The main aim was to provide as much information as possible concerning data used for the methodology and results sections. However, we cut the following parts, trying to reduce the introduction.

Lines 58-66

Lines 85-91

Lines 114-119

The manuscript is only about the engine. Thus, this should reflect in the title.

The paper is about a generic modelling of a retrofitting cost estimation. The application is mainly focused on engine and OBS retrofitting. However, within AGILE4.0 project also other retrofitting packages have been considered. That is the reason why in the title we mentioned retrofitting costs modelling.

The authors should add a paragraph to the second chapter that retrofitting of an entire engine is not the only way how to reduce the noise emission. There are plenty of other options, like adding chevrons (Rask et al. (2011) "Understanding how chevrons modify noise in supersonic jet with flight effects") or fluidic injection (Semlitsch et al. (2019) "Transforming the shock pattern of supersonic jets using fluidic injection") having the advantage to be only appied during take-off and approach.

References added in line 211.

Please check your equations. In Eq. 5, the authors write Np_k and Oh_k but they are described as N_pk and O_hk. See also line 407.

Coherently modified.

There are some details that feel questionable. Like how do you want to mount a BPR 15 engine on a regional jet?

For each engine upgrade solution (so for each engine BPR) we evaluated the engine geometry and dimensions through GasTurb software. From these data, the clearance has been accounted as design constraint. All the engines accounted for the analyses comply with this constraint. I agree with you that it is not easy to install a BPR 15 engine on a regional jet aircraft. For simplicity, we decided to show more in depth only the AEA BPR 15 solution results, which represent the most technologically developed retrofit option. However, AEA BPR 12 solution results are very close to the one shown for AEA BPR 15 one (as possible to see from Figure 9, Figure 10 and Figure 15), by consequence the conclusion addressed at the end of the paper can be equally applied also for a more feasible solution.

There are some minor language errors. For example, "to show a costs-benefits analysis" -> "to show a cost-benefit analysis".

Modified.

In line 213: "However, the engine price is not a factor that really influences the engine choose, ..." -> choice

Modified.

In line 216: there is an additional dot

Modified.

In line 239, "Beltramo et alii [20]" there some "i" too much. Same errors can be found several times in the manuscript: "The reports developed by Beltramo et alii [20] and Large 385 at alii [23] ...". I guess Large 385 is not a name ... .

Replaced “et alii” with “et al.”

Replaced “Large 385” with “Large”. It was a misprint.

In lin 723: Why is Effective captitalised?

Modified.

 '... for seven different retrofittingS' or '... for seven different retrofits'

Replaced with “seven different retrofits”.